# Neural Rule-Execution Tracking Machine For Transformer-Based Text Generation

**Yufei Wang** [1]*, **Can Xu** [2], **Huang Hu** [2], **Chongyang Tao** [2], **Stephen Wan** [3], **Mark Dras** [1],
**Mark Johnson** [1], **Daxin Jiang** [2]†

Macquarie University, Sydney, Australia[1]
Microsoft Corporation, Beijing, China[2]
CSIRO Data61, Sydney, Australia[3]
`yufei.wang@students.mq.edu.au, {mark.dras,mark.johnson}@mq.edu.au`
`{caxu,huahu,chongyang.tao,djiang}@microsoft.com`
`stephen.wan@data61.csiro.au`

## Abstract

Sequence-to-Sequence (Seq2Seq) neural text generation models, especially the pre-trained ones (e.g., BART and T5), have exhibited compelling performance on various natural language generation tasks. However, the black-box nature of these models limits their application in tasks where specific rules (e.g., controllable constraints, prior knowledge) need to be executed. Previous works either design specific model structures (e.g., Copy Mechanism corresponding to the rule "the generated output should include certain words in the source input") or implement specialized inference algorithms (e.g., Constrained Beam Search) to execute particular rules through the text generation. These methods require the careful design case-by-case and are difficult to support multiple rules concurrently. In this paper, we propose a novel module named Neural Rule-Execution Tracking Machine (`NRETM`) that can be equipped into various transformer-based generators to leverage multiple rules simultaneously to guide the neural generation model for superior generation performance in an unified and scalable way. Extensive experiments on several benchmarks verify the effectiveness of our proposed model in both controllable and general text generation tasks.

## 1 Introduction

Transformer-based neural language models (LMs), such as GPT/BART [1–3], have led a wave of new trends in natural language generation, producing texts of prominent quality. They are trained roughly on huge amounts of text corpora to reconstruct the full sentences (i.e., next coming tokens and missing text fragments). Despite their success in varieties of NLP tasks, we argue that the black-box nature of these models leads to inefficiently learning to follow constraints and incorporating prior knowledge.

In controllable text generation, most relevant studies [4–6] focus on controlling high-level text attributes (e.g., topic, sentiment) or simply keyword/phrase. More complex fine-grained control constraints such as "generate a sequence of tokens with 'apple' in the first sentence which has 15 words and 'orange' or 'oranges' in the fourth sentence" are less explored. A very recent work [7] reveals that large-scale LMs do not learn to obey the underlying constraints reliably, even in a quite simple constrained generation task (cover all the given keywords without hallucinating new ones). In general text generation, existing works on various tasks reveal the benefit of incorporating task-specific prior knowledge: machine translation [8] (e.g., each source phrase should be translated

---

*Work done during the internship at Microsoft STCA.

†Corresponding author: Daxin Jiang (djiang@microsoft.com).

into exactly one target phrase), text summarization [9] (e.g., the lead bias: front loading the most salient information), dialogue generation [10] (e.g., humans tend to repeat entity names or even long phrases in conversation). However, they either need designing specific model architectures (e.g., Coverage Mechanism and Copy Mechanism) or devising well-designed learning objectives (e.g., GSG [11]). These methods require careful design case-by-case and are difficult to combine multiple arbitrary constraints or prior knowledge simultaneously.

Motivated by the above research dilemma, we take the *first step* towards building an unified framework to handle *Fine-grained Control* and *Prior Knowledge Integration* and propose a novel module *Neural Rule-Execution Tracking Machine* (NRETM) [3] Specifically, NRETM is a trainable neural module that can be equipped with transformer-based sequence-to-sequence pre-trained LMs. It can handle constraints in any *Predicate Logic Formula*, which crucially includes the arbitrarily complicated relations among different control tasks. For example, the above fine-grained constraint can be written as:

$$\big(InSen(apple, 1) \wedge Len(1, 15)\big) \wedge \big(InSen(orange, 4) \vee InSen(oranges, 4)\big)$$

To build NRETM, we combat three major challenges: *i)* modeling the complicated relationships among control tasks and the logic operators (i.e., $\wedge$, $\vee$) in the constraint expressions; *ii)* an unified control system is required to execute different control tasks simultaneously and *iii)*, the control signals for different control tasks should be properly aligned with the constraint expressions. NRETM uses the encoder of transformer-based pre-trained sequence-to-sequence LMs to model the relationship between control tasks and the logic operators. NRETM completes different control tasks via non-differential *Logic Trackers* (empowered by executable programs) in an unified control progress system during the decoding process. Finally, the encoded constraint expressions and control progress signals are combined together in the transformer decoder. NRETM is fine-tuned with the pre-trained LMs (except logical trackers) to follow the control progress signal and predicate logic formula. NRETM reconciles symbolic computing (that has precise logic and numerical calculation capabilities from logic trackers) with neural language generation (that has an exceptional ability of wording and phrasing), which results in both the accurate controllability and the superior generation performance.

For evaluation, we select three representative benchmarks because all of them involve constraints or prior knowledge, allowing us to verify the effectiveness of our proposed NRETM model: ROCStories [12] are five-sentence stories with complicated predicate constraints over the story structure; Commonsense Generation task [13] with the constraints of mentioning all input concepts; TED15 Zh-En document-level machine translation benchmark [14] with prior knowledge of translating input sentences one by one.

Our contributions in this work are three-fold: (1) To the best of our knowledge, we are the first to propose a general framework that incorporates control signal and prior knowledge, formulated as predicate logic constraints, into transformer-based seq2seq text generation models; (2) We train (or fine-tune) the transformer-based seq2seq text generation models to follow the predicate logic constraints(i.e., control signal or prior knowledge) by dynamically updating the rule execution intermediate progress value to the text decoder; and (3) Empirical verification of the effectiveness of the proposed approach on three benchmarks.

## 2 Approach

This section first formalizes *fine-grained content control* task, then introduces an overview of proposed NRETM model, followed by diving into details of each component.

### 2.1 Fine-Grained Content Control

In this work, we focus on *fine-grained content control* task where the model input consists of predicate logic constraints $\mathbf{x} = [x_1, \ldots, x_{l_x}] \in \mathcal{X}$ that should be satisfied in the outputs and optional context input $\mathbf{c} = [c_1, \ldots, c_{l_c}]$. The encoder takes concatenation of $\mathbf{x}$ and $\mathbf{c}$ (i.e., $[\mathbf{c}; \mathbf{x}]$) as input. At decoding step $t$, the decoder take $\mathbf{y}_{:t} = [y_1, \cdots, y_t] \in \mathcal{Y}$ as input and generate $\mathbf{y}_{t+1}$.

---

[3]Our Source Code can be found in `https://github.com/GaryYufei/NRETM`

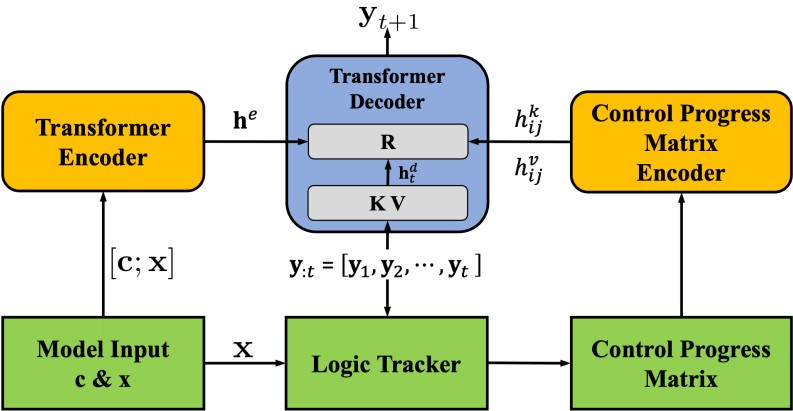

Figure 1: An overview of NRETM. The rounded-angle boxes in the upper row are trainable neural components and the right-angle boxes in the lower row are non-differentiable symbolic components. The predicate logic constraints are modeled as follows: 1) the transformer encoder handles the relationships among the predicates and basic logic operators; 2) *Logic Tracker* keeps track of the control progress of all predicates simultaneously; 3) the encoded expression and control progress are combined in the transformer decoder to guide NRETM to satisfy the constraints.

## 2.2 Predicate Logic Constraint

We define predicate $U(\mathbf{a}, \mathbf{y})$ as a boolean function that indicates whether output $\mathbf{y}$ has satisfied control task $\mathbf{a}$ which could be values (e.g., status, total length, stop word counts) or lexicons (e.g., copying particular words/phrases). In this paper, NRETM accepts predicate logic constraints in Conjunctive Normal Form (CNF): $(U_1 \cdots \vee U_i) \wedge \cdots \wedge (U_k \cdots \vee U_n)$. Each predicate logic constraint includes multiple predicates $U_i$ and basic logic operators (e.g., $\vee$, $\wedge$ and brackets).

## 2.3 Neural Rule-Execution Tracking Machine

NRETM can be equipped into transformer-based sequence-to-sequence LMs. Figure 1 illustrates an overview of our neural rule-execution tracking machine (NRETM). To enable LMs to follow predicate logic constraints, it is essential to 1) model the complicated relationships among predicates and basic logic operators; 2) control multiple predicates (i.e., control tasks) in the constraints simultaneously; 3) combine the control signals with the predicate logic constraint expressions. For 1), we treat the whole constraint expressions as natural language sentences and feed it into the transformer encoder. For 2), we propose a set of unified control signals that can be used to dynamically describe the step-wise execution progress of different predicates. For 3), we represent the control signals as relative position embedding and align them with encoded constraints expressions in the transformer decoder.

### 2.3.1 Encoding Predicate Logic Constraints

Given predicate logic constraint expression $\mathbf{x} = [x_1, \ldots, x_{l_x}]$ where $x_i$ either corresponds to a predicate $U_i$ or a basic logic operator, we feed $\mathbf{x}$ into the transformer encoder. Due to the tokenization strategies of pre-trained LMs, each $x_i$ may be tokenized into a continuous token sequence. $\mathbf{x}$ is tokenized into $\mathbf{t} = [t_1, \cdots, t_{l_t}]$ where $l_t \geq l_x$ and there exists one-to-one mapping $\mathrm{m}(t_i) = x_j$. We use $\mathbf{h}^e$ to denote the encoder output of $\mathbf{x}$. As pre-trained LMs is trained with significant amount of natural language sentences, it should encode complicated sequential relationships within the constraints expressions.

### 2.3.2 Mentoring Control Progress

Specialized controlling components (e.g., Constrained Beam Search [15] and Copy Mechanism [10]) can only be used for limited control tasks. To enable unified controlling system, we propose to complete control by *mentoring control progress*. We describe the control progress of different predicates using an unified progress state system. Each predicate $U_i$ has a corresponding *Logic Tracker* $Q_{U_i}(\mathbf{y})$, which is a non-differentiable executable program (i.e., written by Python) and takes

| **Control Progress Matrix** | | | | | | | | | | |
|---|---|---|---|---|---|---|---|---|---|---|
| **x**    **y**:t | It | is | raining | . | The | woman | cannot | drive | her | car | EOS |
| **Copy(car)** | S2 | S2 | S2 | S2 | S2 | S2 | S2 | S2 | S2 | S2 | S3 |
| **&** | S0 | S0 | S0 | S0 | S0 | S0 | S0 | S0 | S0 | S0 | S0 |
| **SWC(5)** | S2, 5 | S2, 4 | S2, 3 | S2, 3 | S2, 3 | S2, 2 | S2, 2 | S2, 1 | S2, 1 | S3 | S3 |
| **&** | S0 | S0 | S0 | S0 | S0 | S0 | S0 | S0 | S0 | S0 | S0 |
| **Length(2, 6)** | S1 | S1 | S1 | S1 | S2, 6 | S2, 5 | S2, 4 | S2, 3 | S2, 2 | S2, 1 | S3 |

Figure 2: A running example of our NRETM model with three logic constraints (i.e., Copy(car) & SWC(5) & Length(2,9); The output should include "car" and 5 stop words. The length of second sentence should be 6. The words are stop words.

current generated outputs and returns one progress state at each generation step, formulated as follows:

$$Q_{U_i}(\mathbf{y}) = \begin{cases} \text{S0} & U_i \text{ is } \varnothing \\ \text{S1} & U_i \text{ is not triggered in } \mathbf{y} \\ (\text{S2,V}) & U_i \text{ is in progress in } \mathbf{y} \\ \text{S3} & U_i \text{ is satisfied in } \mathbf{y} \end{cases} \tag{1}$$

where State S0 always is assigned to non-predicate $\varnothing$ (i.e., basic logic operators in the constraint expression); State S1 means the tracking for predicate $U_i$ is not triggered in $\mathbf{y}$. For example, when controlling the stop word counts of the second sentence, the *Logic Tracker* returns S1 when the LMs are generating the first sentence; State S2 means predicate $U_i$ is in progress and $V$ is the optimal intermediate value that allows fine-grained tracking. For example, in generation length control, $V$ could be total target length minus the current length informing pre-trained LMs the number of words left to satisfy the constraint; State S3 means $U_i$ is satisfied in $\mathbf{y}$. In short, *Logic Tracker* unifies different predicates by returning the same set of control signals.

**Global Or-Clause Update:** Each *Logic Tracker* traces the execution progress of its corresponding predicate $U_i$ independently. This independent tracing strategy works well in the And-Clause because all involved predicates are required to reach State S3. However, only a subset of predicates are required to reach State S3 in the Or-Clause. Our preliminary experiment shows that the independent tracing strategy trains the model not to complete the constraints. To solve this issue, we propose to update the status of all predicates in the same Or-Clause to State S3 when one of the predicates reach State S3. This forces all predicates finish themselves in State S3 and improves the constraint satisfaction ratio in the Or-Clause.

**Control Progress Matrix:** Given the predicate logic constraint expressions $\mathbf{t} = [t_1 \cdots, t_{l_t}]$, we further define *Control Progress Matrix $\mathcal{S}$* to align the predicates with their control progress signals returned by *Logic Trackers*:

$$\mathcal{S} = [\text{C}(\mathbf{t}, \varepsilon); \text{C}(\mathbf{t}, \mathbf{y}_{:1}); \cdots ; \text{C}(\mathbf{t}, \mathbf{y}_{:t})] \tag{2}$$

$$\text{C}(\mathbf{t}, \mathbf{y}_{:t}) = [\text{v}(t_1, \mathbf{y}_{:t}), \cdots , \text{v}(t_{l_t}, \mathbf{y}_{:t})] \tag{3}$$

where $\varepsilon$ is the empty string at first decoding step. $\mathcal{S}$ is a two-dimensional matrix where each row describes the control progress of all tokens in $\mathbf{t}$ at a single decoding step and each column describes the control progress of a single token in $\mathbf{t}$ along all decoding steps. Recall that basic logic operators in predicate logic constraint expressions do not require control progress tracking. Each cell $\mathcal{S}_{i,j}$ in $\mathcal{S}$ is formulated as:

$$\mathcal{S}_{i,j} = \text{v}(t_i, \mathbf{y}_{:j-1}) = \begin{cases} Q_\varnothing(\mathbf{y}) & \text{m}(t_i) = x_k \text{ and } x_k \text{ is a basic logic operator} \\ Q_{U_q}(\mathbf{y}) & \text{m}(t_i) = x_k \text{ and } x_k \text{ is a predicate } U_q \end{cases} \tag{4}$$

**Example:** In Figure 2, we are given three logic constraints, *a)* copy "car"; *b)* the stop word ratio of the output should be 0.5 and *c)* the length of second sentence should be 6. The basic logic operators & are assigned with S0. Length control and Stop Word Ratio maintain intermediate values (e.g., the residual Length and Stop Word Ratio). The length control is assigned with S1 when generating the first sentence because it will only be triggered in the second sentence. Copy control does not have intermediate values and its State are updated from S2 to S3 only when the corresponding words (at step 10 in our example) appear in the $\mathbf{y}_{:t}$.

**Control Progress Matrix Encoder:** *Control Progress Matrix* $\mathcal{S}$ aligns the results from *Logic Tracker* with the encoded predicate logic constraint expressions. However, $\mathcal{S}$ is a non-differentiable symbolic matrix with each cell $\mathcal{S}_{i,j}$ being discrete symbol S0 to S3 combined with additional numbers (i.e., $V$). As the encoder has already captured the inter-relationship in the predicate logic constraints, we only model each cell $\mathcal{S}_{i,j}$ independently. To support various types of predicates, we treat $\mathcal{S}_{i,j}$ as a string and encode it using a single-layer transformer-based encoder ShallowEncoder which shares the same vocabulary and word embeddings as the pre-trained LMs:

$$\mathbf{h}_{ij}^s = \text{ShallowEncoder}(\mathcal{S}_{i,j}) \tag{5}$$

$$\bar{\mathbf{h}}_{ij}^s = \text{MeanPooling}(\mathbf{h}_{ij}^s) \tag{6}$$

where $\mathbf{h}_{ij}^s \in \mathbb{R}^{l_{ij}^s \times d}$, $\bar{\mathbf{h}}_{ij}^s \in \mathbb{R}^d$ and $l_{ij}^s$ is the length of the tokenized $\mathcal{S}_{i,j}$ and $d$ is the hidden size of ShallowEncoder. We use $\bar{\mathbf{h}}^s$ to denote the neural representation of whole $\mathcal{S}$.

### 2.3.3 Combining Predicate Logic Constraint with Control Progress Matrix

Finally, we combine the encoded *Predicate Logic Constraints* $\mathbf{h}^e$ with the encoded *Control Progress Matrix* $\bar{\mathbf{h}}^s$ in the transformer-based pre-trained LMs. Injecting $\bar{\mathbf{h}}^s$ into the transformer encoder would result in encoder content re-computation at each decoding step and stop the standard parallel training for transformer-based decoders. In addition, as *Control Progress Matrix* incrementally increases as the decoding goes on, it is reasonable to equip $\bar{\mathbf{h}}^s$ into the transformer decoder. Given the encoder output $\mathbf{h}^e$, decoder input $\mathbf{y}_{:t}$, the probability of the next token $y_{t+1}$ can be calculated by:

$$\mathbf{h}_t^d = \text{KV}(\mathbf{W}_q^s \mathbf{y}_{:t}, \mathbf{W}_k^s \mathbf{y}_{:t}, \mathbf{W}_v^s \mathbf{y}_{:t} \tag{7}$$

$$o_{t+1} = \text{CrossKV}(\mathbf{W}_q^c \mathbf{h}_t^d, \mathbf{W}_k^c \mathbf{h}^e, \mathbf{W}_v^c \mathbf{h}^e) \tag{8}$$

$$p(y_{t+1}|x_{1:l_x}, y_{1:t}) = \text{softmax}(\mathbf{W}_o \, o_{t+1}) \tag{9}$$

where $o_{t+1} \in \mathbb{R}^{d_c}$ is the hidden state at step $t$ with $d_c$ the hidden size, and $\mathbf{W}_o \in \mathbb{R}^{|V| \times d_c}$, Both KV and CrossKV are the standard key-value self-attention described in [16]. In the CrossKV which takes $\mathbf{h}_t^d$ and $\mathbf{h}^e$ as input, the resulting attention score matrix has the same size as $\mathcal{S}$, making CrossKV suitable to incorporate our *Control Progress Matrix*.

**Control Progress Matrix as Relative Position:** Inspired by [17] which incorporates token relative positions into the self-attention module, we propose to inject Control Progress Matrix as the "relative positions" between encoder output $\mathbf{h}^e$ and current decoder input $\mathbf{y}_{:t}$ in the cross-attention (Eq. 8) module. Following this approach, we linearly project each $\bar{\mathbf{h}}_{ij}$ into *Control Progress Matrix key* $\mathbf{h}_{ij}^k = \mathbf{W}_k^f \cdot \bar{\mathbf{h}}_{ij}^s + \mathbf{b}_k^f$ and *Control Progress Matrix Value* $\mathbf{h}_{ij}^v = \mathbf{W}_v^f \cdot \bar{\mathbf{h}}_{ij}^s + \mathbf{b}_v^f$. All transformer decoder layers share the same representations. Eq. 8 is changed to:

$$o_{t+1} = \text{R}(\mathbf{W}_q^c \mathbf{H}_t^d, \mathbf{W}_k^c \mathbf{H}^e, \mathbf{W}_v^c \mathbf{H}^e, \mathbf{h}^k, \mathbf{h}^v) \tag{10}$$

where $\mathbb{R}^{l_x \times t \times d}$ and $R$ is the Self-Attention function with relative position, defined as follows:

$$R(\mathbf{q}, \mathbf{k}, \mathbf{v}, \mathbf{m}^k, \mathbf{m}^v)_j = \sum_{i=1}^{l_x} \mathbf{a}_{i,j}(\mathbf{v}_i + \mathbf{m}_{i,j}^v) \tag{11}$$

where $\mathbf{a}_{*,j} = Softmax(\mathbf{e}_{*,j})$ and $\mathbf{e}_{i,j} = \mathbf{q}_j(\mathbf{k}_i + \mathbf{m}_{i,j}^k)^T d^{-1/2}$.

### 2.4 Why NRETM Could Satisfy Constraints

A powerful implicit compulsion comes from the combined force of two aspects: 1) before generating the EOS token (i.e., End-Of-Sequence Token), all the predicate constraints should be satisfied. As demonstrated in Fig 2, all elements in *Control Progress Matrix* are set to "satisfied" (i.e., S3) at EOS position; 2) The pre-trained LMs are trained to generate text with limited length. Such a soft way of combining symbolic operators (good at logical and mathematical calculations ) and neural operators (good at wording and phrasing) can retain their respective strengths to the utmost extent.

### 2.5 What If NRETM Fails to Satisfy Constraints

NRETM does not forces the pre-trained LMs to execute the hard constraints on the text decoder explicitly, but instead, provides *Control Progress Matrix* as input features describing rule execution

intermediate values to the text decoder. That is, no explicit effect when NRETM fails to satisfy the constraints. It is possible that our text generators decide to stop the generation before completing all constraints. In our experiments, NRETM has less than 1% chance not to complete all constraints.

## 2.6 The Generalization Ability of NRETM

The generalization ability of NRETM comes from two aspects: 1) NRETM can construct new constraints via combining pre-trained predicates with basic logic operators in arbitrarily complicated ways; 2) To expand a new predicate, users only need to implement the corresponding *Logic Trackers*, which returns S1-S3 and intermediate values, via executable programs.

## 3 Experiment

We test our proposed NRETM on the controllable text generation and general text generation tasks. For controllable text generation, we verify NRETM on the complex fine-grained control instructions in the ROCStories Benchmark [12]. Further, we test NRETM on the general text generation tasks, commonsense generation and document-level machine translation, to show that NRETM can efficiently integrate prior knowledge into seq2seq models towards superior generation performance.

Table 1: Controllable ROCStories Experiment Results.

| Predicate Logic Constraint | M | CSR | RL | BS | B1 | B2 |
|---|---|---|---|---|---|---|
| $\wedge_{i=1}^{4}\big(\mathrm{InSen}(w_i, y^{p_i})\big)$ | T5 | 94.6 | 56.1 | 91.7 | 52.5 | 27.7 |
| | NRETM | 97.6 | 56.0 | 91.7 | 52.1 | 27.5 |
| $\mathrm{Copy}(w_1) \wedge_{i=1}^{3}\big(\mathrm{Order}(w_i, w_{i+1})\big)$ | T5 | 95.6 | 55.5 | 91.5 | 51.4 | 26.5 |
| | NRETM | 98.3 | 55.6 | 91.4 | 47.5 | 25.0 |
| $\wedge_{i=1}^{2}\big(\mathrm{InSen}(w_i, y^{p_i}) \wedge \mathrm{Len}(y^{p_i}, l_{p_i})\big)$ | T5 | 15.0 | 33.0 | 87.8 | 38.6 | 11.5 |
| | NRETM | 78.8 | 33.1 | 87.8 | 38.4 | 11.5 |
| $\mathrm{InSen}(w_1, y^{p_1}) \wedge \mathrm{InSen}(w_2, y^{p_2}) \wedge (\neg\mathrm{InSen}(w_3, y^{p_2}))$ | T5 | 32.4 | 33.2 | 87.8 | 36.1 | 11.8 |
| $\wedge(\mathrm{Len}(y^{p_1}, l_{p_1}) \vee \mathrm{SWC}(y^{p_1}, s_{p_1}))$ | NRETM | 70.0 | 32.7 | 87.7 | 36.9 | 11.7 |
| $\wedge_{i=1}^{2}\big(\mathrm{InSen}(w_i, y^{p_i}) \wedge (\mathrm{Len}(y^{p_i}, l_{p_i}) \vee \mathrm{SWC}(y^{p_i}, s_{p_i}))\big)$ | T5 | 18.7 | 33.2 | 87.9 | 37.6 | 11.5 |
| | NRETM | 64.7 | 33.2 | 87.9 | 38.5 | 11.7 |

### 3.1 Controllable ROC Stories

ROCStories is a corpus of five-sentence stories that capture a rich set of causal and temporal commonsense relations between daily events. Following [18], we extract key phrases from the ground-truth stories. In this experiment, we design multiple predicate logic constraints to inform NRETM about the stories to be generated and verify if NRETM can follow these constraints exactly.

**Predicate Logic Formulation** As shown in table 1, five constraints with increasing difficulties are used: (1) Generate a story with storyline $w_i$ in the $p_i{}^{th}$ sentence. (2) Generate a story with an ordered storyline $w_1, \cdots, w_4$ (3) Generate a story with storyline $w_i$ in the $p_i{}^{th}$ sentence which has $l_{p_i}$ words ($i = 1, 2$). (4) Generate a storyline $w_1$ in the $p_1{}^{th}$ sentence which has $l_{p_1}$ words or $s_{p_1}$ stop words and $w_2$ in the $p_2{}^{th}$ sentence that does not mention $w_3$ (5) Generate a storyline $w_i$ in the $p_i{}^{th}$ sentence which has $l_{p_i}$ words or $s_{p_i}$ stop words ($i = 1, 2$).

**Baselines and Metrics** Both baseline and NRETM use T5-Base model [19]. We report *Constraints Success Ratio* (CSR), the ratio of stories that completely satisfy the given constraints. We additionally report ROUGE-L (RL), BERT-Score (BS), BLEU-1/4 (B1/4) to show the generated stories quality.

**Main Results** As shown in Table 1, in all five predicate logic constraints, compared to the T5 model, the NRETM model achieves higher Constraint Success Ratio and maintains a similar level of ROUGH-L, showing that the NRETM model can be flexibly controlled without loss of generated text quality.

Table 2: Experiment Results on Commonsense.

| Method | BS | B1 | B4 | C | S | Constraint | | |
| --- | --- | --- | --- | --- | --- | --- | --- | --- |
| | | | | | | Seen | Novel | ALL |
| T5-Base | 94.5 | 71.3 | 29.2 | 159.4 | 31.9 | 92.9 | 90.1 | 92.7 |
| T5-Base + NRETM $P_c$ | 94.6 | 72.5 | 30.3 | 163.8 | 32.4 | 94.6 | 93.6 | 94.6 |
| T5-Base + NRETM $\hat{P}_c$ | 94.5 | 74.2 | 29.3 | 167.7 | 33.2 | 99.4 | 99.6 | 99.5 |
| T5-Large | 94.8 | 73.0 | 32.4 | 170.3 | 33.1 | 94.8 | 92.4 | 94.6 |
| T5-Large + NRETM $P_c$ | 94.8 | 74.3 | 32.1 | 173.4 | 33.5 | 97.8 | 96.9 | 97.8 |
| T5-Large + NRETM $\hat{P}_c$ | 94.8 | 74.8 | 32.6 | 175.3 | 34.3 | 99.2 | 99.0 | 99.2 |
| T5-Base + G | 92.8 | 58.6 | 40.2 | 110.7 | 27.8 | 100 | 100 | 100 |
| T5-Large + NEUROLOGIC | 94.8 | 73.2 | 32.3 | 169.7 | 32.3 | 99.1 | 98.8 | 99.0 |
| KGBART [23] | - | - | - | 168.3 | 32.7 | - | - | 98.6 |

The gap in CSR between the T5 and NRETM model is moderate in the first two constraints with simple token permutations. However, the success ratio of T5 model drops significantly given constraints that requires long-range numerical tracking (e.g., sentence length and the count of stop words).

## 3.2 Commonsense Generation

COMMONGEN is a generation benchmark dataset target explicitly test machines for the ability of generative commonsense reasoning. Given a set of common concepts the task is to generate a coherent sentence describing an everyday scenario using these concepts.

**Predicate Logic Formulation** The input is an unordered set of $n$ concepts $\mathbf{x} = \{x_i\}_{i=1}^n$. From the expectation of COMMONGEN, one easily obtained prior knowledge is that each $x_i$ must appear in output $\mathbf{y}$. The corresponding predicate logic constraint $P_c$ is:

$$P_c = \wedge_{i=1}^n \big( \text{Copy}(x_i) \big)$$

where $\mathbf{y}$ will appear by default, for the sake of brevity, we have omitted $\mathbf{y}$ in predicate Copy. Another prior knowledge comes from the observation that generating $\mathbf{y}$ requires giving the correct morphological inflections of the concept word rather than copy its original form. Let $\tilde{x}_i = \{\tilde{x}_k^i\}_{k=1}^{|\tilde{x}_i|}$ denote all inflections of $x_i$. $\mathbf{y}$ covers concept $x_i$, if at least one of $\{\tilde{x}_k^i\}_{k=1}^{|\tilde{x}_i|}$ appears. The constraint $\hat{P}_c$ is:

$$\hat{P}_c = \wedge_{i=1}^n \big( \vee_{j=1}^{|\tilde{x}^i|} \text{Copy}(\tilde{x}_j^i) \big)$$

**Baselines and Metrics** We experiment with T5-Base and T5-Large. We equip NRETM into the T5-Large and T5-Base model to incorporate $P_c$ and $\hat{P}_c$ respectively (+ NRETM $P_c$) (+ NRETM $\hat{P}_c$). Grid Beam Search (GBS) [20] (+ G) is a well-designed decoding method that ensures the generation model satisfies the lexical constraints. We only apply GBS to the T5-Base model due to the memory constraint. Following the suggestions in [13], we use CIDEr [21] and SPICE [22] to automatically assess the quality of generated texts. We calculate constraint satisfaction for all constraints (ALL), novel constraints (Novel) and seen constraints (Seen).

**Main Results** Table 2 shows that the NRETM model improves the constraint satisfaction over the baselines for all cases, achieving close to 100% (i.e., 99.5% and 99.2%). While GBS achieves perfect constraint satisfaction (i.e., 100%), doing so significantly degrades the output text quality (more than 50 CIDEr), indicating the necessity integrating prior knowledge in training rather than inference. In addition, both prior knowledge $P_c$ and $\hat{P}_c$ have a positive effect on our model, improving our T5-large baseline by 3.1 and 5.0 CIDEr score, respectively. Finally, our *T5-Large + NRETM $\hat{P}_c$* model outperforms the previous state-of-the-art result [23], which integrates the *ConceptNet* [24] into the BART model, suggesting that our incorporated task-specific prior knowledge could be as powerful as knowledge from large-scale hand-crafted corpus. All of the above shows how potential it is to find a method that could execute multiple rules effectively.

### 3.3 Document-Level Machine Translation

Document-level machine translation tasks is a general text generation task, where the goal is to translate segments of text (up to an entire document). Following [14], we use TED15 Zh-En (from IWSLT 2014 and 2015 [25, 26]) as training and validation set and 2010-2013 TED as the test set.

**Predicate Logic Formulation** The input is an ordered set of $n$ sentences in the source language that form a document $\mathbf{x} = \{x^i\}_{i=1}^n$, the expected output is a translated document $\mathbf{y} = \{y^i\}_{i=1}^n$ in the target language. We observed that neural model is prone to sentence correspondence confusion (the $i^{th}$ sentence in source document is translated as the $j^{th}$ sentence in target document) when doing document-level translation. To alleviate this problem, we propose incorporating Doc-mBART25 with prior knowledge: each source sentence should be translated only once. It is formulated as:

$$\text{TranslatedOnce}(x^i) = \left\{ \begin{array}{ll} S3 & \theta(\mathbf{y}_{:t}) > i \\ S2 & \theta(\mathbf{y}_{:t}) = i \\ S1 & \theta(\mathbf{y}_{:t}) < i \end{array} \right. \tag{12}$$

where $\theta(\cdot)$ returns the line number of $y_t$ in $\mathbf{y}$, as t is monotonic during generation, the status only set to be 2 once. To trace the sentence translation progress, we add an additional End-Of-Sentence token at the end of each sentence to the training data. Once NRETM finishes the $i^{th}$ sentence (generating an end-of-sentence token) in the decoder, we assume that the $i^{th}$ sentence in the encoder has been translated. The predicate logic constraint $P_c$ of this task can be formulated as:

$$P_c = \wedge_{i=1}^n \big( \text{TranslatedOnce}(x^i) \big)$$

**Baselines and Metrics** We combine our NRETM $P_c$ component with the Doc-mBART25 model proposed in [3] which is a state-of-the-art multilingual pre-trained language model. We compare this model with the state-of-the-art non-pretraining and pretraining approaches, including HAN (Hierarchical Attention Networks) [14], Doc-mBART25 and Sen-mBART25 proposed in [3]. When implementing our model, we use the same pre-processing method, blocks segmentation strategy and beam search setting as [3]. TED15 Zh-En provides sentence-to-sentence translation from Chinese to English. We use both document-level (d-BLEU) and sentence-level (s-BLEU) to measure the similarities between generated target document and the source document. We also report Sentence Aligned Ratio (SAR), the ratio of source and target documents with the same sentence count, to show the effectiveness of our control over this translation prior knowledge.

**Main Results** Table 3 shows that the NRETM $P_c$ component helps the Doc-mBART25 model to better capture the sentence-level corresponding relationship between the source and target documents. In particular, sentence-level alignment ratio is improved from 98.7% to 100%. The improvement in s-BLEU (+ 1.1 BLEU) also confirms that our final Doc-mBART25 + NRETM $P_c$ model learns to translate sentences based on the sentence order in source documents.

Table 3: Model Performance on TED15 Zh-En Test.

| Model | s-BLEU | d-BLEU | SAR |
|---|---|---|---|
| Doc-mBART25 + NRETM $P_c$ | 24.9 | 30.4 | 100 % |
| Doc-mBART25 [3] | 23.8 | 29.6 | 98.7 % |
| Sen-mBART25 [3] | - | 28.4 | - |
| HAN [14] | - | 24.0 | - |

### 3.4 Discussion

**Updating Progress in Encoder** In Sec 2.3.3, we incorporate the *Control Progress Matrix* as relative position embeddings in the decoder. To show the importance of this design choice, we conduct an ablation study in Table 4 where the row of *Control Progress Matrix* is concatenated with the encoder output. We find that updating the rule execution progress information with the encoder output contributes little to improve the CSR. This shows that simply extracting rule execution intermediate values is not enough. This could be because the encoder that encodes the rule execution intermediate values cannot effectively broadcast this information into text decoders.

NRETM **Robustness** The above experiment results are based on the perfect training data. In this section, we explore the effect of training data noise. We corrupt the training data by replacing the input commonsense keywords with a random sampled one under the probability 5%, 10%, 15%, 25%, and 50% (Validation and Test Split remain unchanged). As shown in Table 5, in all noise levels,

Table 4: Updating Rule Execution Progress with Encoder Output

| Constraint | (1) | | (2) | | (3) | | (4) | | (5) | |
|---|---|---|---|---|---|---|---|---|---|---|
| Model | RL | CSR | RL | CSR | RL | CSR | RL | CSR | RL | CSR |
| T5 | 56.1 | 94.6 | 55.5 | 95.6 | 33.0 | 15.0 | 33.2 | 32.4 | 33.2 | 18.7 |
| NRETM | 56.0 | 97.6 | 55.6 | 98.3 | 33.1 | 78.8 | 32.7 | 70.0 | 33.2 | 64.7 |
| enc-update | 56.0 | 96.8 | 55.6 | 97.2 | 33.0 | 14.2 | 32.7 | 28.5 | 33.3 | 16.8 |

NRETM successfully achieves higher constraint coverage (i.e, Cons) and CIDEr score than the T5 baseline model, showing that NRETM is robust to the training data noise. It is worthwhile to note that the main goal of NRETM is to incorporate constraints that are satisfied by the training data into transformer-based seq2seq text generators. It is reasonable to assume that in practice, the noise level should be relatively low (e.g., 0% - 10%).

Table 5: NRETM performance on Commonsense Test Split under different noise levels.

| Noise | 0% | | 5% | | 10% | | 15% | | 25% | | 50% | |
|---|---|---|---|---|---|---|---|---|---|---|---|---|
| Model | C | Cons | C | Cons | C | Cons | C | Cons | C | Cons | C | Cons |
| T5 | 170.3 | 94.6 | 168.8 | 93.3 | 168.0 | 93.7 | 163.8 | 92.9 | 160.7 | 91.3 | 102.0 | 66.7 |
| NRETM | 175.3 | 99.2 | 169.5 | 96.5 | 168.5 | 96.2 | 167.1 | 94.6 | 161.2 | 92.4 | 149.7 | 87.4 |

**Zero-Shot Execution** In Table 1, we show that the pre-trained language model T5 cannot handle complicated and fine-grained constraints even after fine-tuning. Here, we further demonstrate that NRETM model is capable to handle zero-shot rule execution. We train the T5 and NRETM model to only mention keywords in the $3^{rd}$, $4^{th}$ and $5^{th}$ sentence and test these models to mention keywords in the first and second sentence of the whole story. As shown in Table 6, although both T5 and NRETM model mention most of the keywords (95.7% and 98.3% respectively) in the generated story, the T5 model only mention 19.7% of keywords

Table 6: Novel Sentence Index Experiment. MR for Mention Ratio.

| model | CSA | MR | RL |
|---|---|---|---|
| T5 | 19.7 | 95.7 | 30.0 |
| NRETM | 97.7 | 98.3 | 33.3 |

in the correct sentence and the NRETM model makes 97.7% of keywords correct. This is becuase the T5 model cannot recognize the novel sentence index (i.e., the first and second) during the generation. The logic tracker helps the NRETM model to generalize to handle these cases.

**Running Efficiency** We compare the inference time (in minutes) for NRETM on the test split of commonsense generation task in Table 7. All models use the beam search decoding algorithm with beam size 5. Adding NRETM components to T5-Base and T5-Large approximately double the inference time. While the Grid Beam Search (GBS) algorithm uses a much longer inference time. Compared to existing constrained decoding approaches, NRETM uses much less computational costs.

Table 7: Inference Time On Commonsense Generation Task Test Split (in minutes).

| Model | Baseline | +NRETM | +GBS |
|---|---|---|---|
| T5-Base | 1.05 | 2.27 | 84 |
| T5-Large | 1.32 | 2.6 | - |

# 4 Related Work

NRETM is mainly related to two lines of research work in text generation: constrained decoding and prior knowledge integration.

**Constrained Decoding** NEUROLOGICEarly work in constrained decoding can be traced back to dual decomposition and lagrangian relaxation [27, 28]. These works focus on sequence labelling and parsing problems where the solution space is relatively small, compared to text generation tasks. Research efforts in text generation tasks [29–31] involve controllable generation methods where the generators are trained on text data with the labeled target attributes. CTRL [4], PPLM [6] and CoCon [32] are recent approaches that built on the transformer-based large-scale pretrained LMs,

they pay more attention on controlling high-level attributes, phrases and keywords. [33, 34] propose to trace the control task progress in the text generation decoder. [33] treats the control signal as training loss in memory network and [34] treats the control signal as additional input features. [35] controls the text generation outputs via mentoring the output gradient. However, these work only focus on specific controlling tasks such as phrases copying and generation length. While NRETM focuses on controlling text generation to follow arbitrary logical constraints, leading to a fine-grained control. They can be seen as special cases of NRETM. Recently, GDC [36] permits to specify both pointwise and distributional constraints over the target LMs. Very recently, NEUROLOGIC [7] was proposed to generate fluent text while satisfying complex lexical constraints (in a predicate logic form). There are three main differences between NRETM and NEUROLOGIC: *1)* NEUROLOGIC only provides control constraints over the text generators. Instead, NRETM is a general framework that provides control constraints (e.g., copy or not copy words) and prior knowledge (e.g., translating sentences one by one). NEUROLOGIC can be viewed as a special case of NRETM; *2)* NEUROLOGIC is an inference-only algorithm that only controls the model to generate or avoid specific words or phrases at decoding time; while NRETM fine-tunes the pre-trained transformer-based seq2seq text generators with the predicate logic constraints; *3)* NEUROLOGIC only supports the "copy" predicate (i.e., to generate or not to generate specific words or phrases), while NRETM is a general framework that supports various control predicates. NRETM supports 6 kinds of logic operators in this paper, and it is also possible for users to expand new logic operators.

**Prior Knowledge Integration**   Existing efforts [37–41] to incorporate prior knowledge into sequence-to-sequence framework either resort to modifying model architectures, including adding external memory components, specialized decoding method or designing training objectives, including minimum risk training. These methods usually can only support to inject one narrow type of knowledge into the neural models. To the best of our knowledge, we first attempt to formalize the prior knowledge integration in seq2seq generation as text generation that conforms to predicate logic constraints.

# 5   Conclusion and Future Work

In this paper, we propose a unified controllable generation framework that leverages predicate logic constraints to implement efficient complex fine-grained control and scalable prior knowledge integration. We explore and compare two controllable strategies: dynamic tracking and static strategy, and show that the proposed dynamic tracking mechanism significantly outperforms the static ones. Empirical results on three benchmarks indicate that NRETM could achieve accurate control and exhibits a superior generation ability over different tasks. Pre-trained models have been the dominant paradigm in natural language processing, and researchers resort to massive data and large-scale models to improve performance. We unify the rules used in various tasks into the form of predicate logic, provide the possibility to pretrain models on massive rules. In the future, we will explore pre-training large-scale neural rule-execution machine with massive rules and data.

# Broader Impact

Our work proposes a unified and scalable approach to efficiently perform fine-grained controllable text generation and incorporate multiple prior knowledge for superior text generation performance. This work uses story generation, machine translation, commonsense generation as applications to verify the effectiveness. However, while our proposed method achieves promise performance on several benchmarks, deployment of our method in the real world requires a careful analysis of potential societal benefits and harms (e.g., the harms associated with furthering negative stereotypes against certain vulnerable groups). The potential ethical issues include: powerful language models might be used to generate abuse, faked or misleading content in the news or on social media; they might pose safety concerns if they are used to generate harassing or hateful materials. In order to mitigate these risks, it is possible to use AI systems to fight against misleading content and harassing material. However, as discussed in previous work [42, 43], mitigating these risks could be an extremely complex socio-technical problem that many are working to understand and solve.

## Acknowledgement

We thank anonymous reviewers for their insightful suggestions to improve this paper.

## Funding Transparency Statement

Yufei Wang, Can Xu, Huang Hu, Chongyang Tao and Daxin Jiang are supported by Microsoft Software Technology Center at Asia (STCA). Yufei Wang also receives a MQ Research Excellence Scholarship and a CSIRO's DATA61 Top-up Scholarship.

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
