# Appendix

In this Appendix, we provide more details and examples for our proposed `NRETM` models. A.1 provides the case study for three evaluation tasks. A.2 provides the definitions of six predicates used in this paper. A.3 provides implementation details of our models.

## 5.1 Case Study

In this case study, we select representative examples from our T5 baseline and `NRETM` models. The *Logic expression* is the actual predicate logic constraints that we use in the model. The *Input expression* is the actual string that we feed into the T5 model simply because T5 vocabulary does not include $\wedge$, $\vee$, and $\neg$.

Table 8: Generated examples of the T5 and `NRETM` model for the story generation task.

| | **Task 1: Story Generation** |
|---|---|
| Input expression: | Order(hated, stupid) & Order(stupid, insulting) & Order(insulting, punched) |
| Logic expression: | Order(hated, stupid) $\wedge$ Order(stupid, insulting) $\wedge$ Order(insulting, punched) |
| **T5**: | I had a crush on a man. I told him I was stupid. He hated me for insulting me. I punched him in the face. He told me i was stupid. |
| **NRETM**: | I had a crush on a guy. He was very rude. He hated me because i thought he was stupid and insulting me. I punched him in the face. He slapped me. |
| Input expression: | InSen(walking around, 2) & InSen(school yard, 2) |
| Logic expression: | InSen(walking around, $y^2$) $\wedge$ InSen(school yard, $y^2$) |
| **T5**: | I went to the park today. I was walking around the school yard. I saw a squirrel in the yard. I saw a squirrel in the yard. I took a picture of the squirrel. |
| **NRETM**: | Sam was playing in the yard. He was walking around in the school yard. He saw a squirrel in the yard. He ran into the squirrel. Sam was able to retrieve the squirrel. |
| Input expression: | InSen(waitressing job, 3) & (Len(3, 16) ‖ StopWordCount(3, 8)) & (not InSen(tenacity, 3)) & InSen(talent agent, 4) |
| Logic expression: | InSen(waitressing job, $y^3$) $\wedge$ (Len($y^3$,16) $\vee$ StopWordCount($y^3$,8)) $\wedge$ ( $\neg$ InSen(tenacity, $y^3$)) $\wedge$ InSen(talent agent, $y^4$) |
| **T5**: | Sarah had a crush on a girl named Sarah. Sarah s crush was Sarah's boyfriend. Sarah's boyfriend had a waitressing job and Sarah had a crush on Sarah . Sarah s boyfriend was a talent agent. Sarah's crush was sarah's boyfriend. |
| **NRETM**: | Sarah wanted to be a chef. but she didn't have the money to make a living. She applied for a waitressing job but she did n't get a job at all . Sarah was approached by a talent agent. Sarah was hired and now she is a chef. |
| Input expression: | InSen(longest grocery trip, 1) & Len(1, 12) & InSen(line forever, 4) & Len(4, 9) |
| Logic expression: | InSen(longest grocery trip, $y^1$) $\wedge$ Len($y^1$, 12) $\wedge$ InSen(line forever, $y^4$) $\wedge$ Len($y^4$, 9) |
| **T5**: | Sam was on the longest grocery trip of his life. He was so hungry he couldn't wait to get to the store. When he got to the store he had to wait in line forever. Sam had to wait in line forever for his next grocery trip. |
| **NRETM**: | I was on the longest grocery trip I ever went on. I had to wait in line for a long time. When I got there the line was long. I had to wait in line forever again. I was so glad I finally got to the store |
| Input expression: | InSen(could buy one, 2) & (Len(2, 13) ‖ StopWordCount(2, 6)) & InSen(would work, 4) & (Len(4, 10) ‖ StopWordCount(4, 7)) |
| Logic expression: | InSen(could buy one, $y^2$) $\wedge$ (Len($y^2$, 13) $\vee$ StopWordCount($y^2$, 6)) $\wedge$ InSen(would work, $y^4$) $\wedge$ (Len($y^4$, 10) $\vee$ StopWordCount($y^4$, 7)) |
| **T5**: | I needed a new computer. I asked my mom if I could buy one. She said yes. I asked her if it would work for me. She said yes |
| **NRETM**: | I needed a new computer. I went to the store to see if I could buy one. I went to the store to buy a computer. I found a computer that would work for me. I bought the computer and was very happy. |

Table 8 shows selected examples in Controllable ROCStories Generation task. This task is to show the controllability of our proposed `NRETM` model. Sentences in red are the ones being controlled. In the first story, the T5 baseline model produces a short sentence and misses the order of storyline "stupid" which should appear after generating the storyline "hated". While our `NRETM` model successfully

completes all storylines in order. In the second story, the NRETM model controls the story generation in a more coherent way than the T5 baseline model. Although both baseline and NRETM model successful incorporate all given storylines, the T5 baseline model inconsistently generates "school yard" just after generating the "park". On the contrary, in the story generated by the NRETM model, Sam consistently stays in the "yard". In the third story, the length and stop word control force the NRETM model to generate sentences with more details, while the T5 baseline simply repeats information from previous sentences. The NRETM model successfully generates eight stop words in the third sentence, whereas the baseline model only generates six stop words (highlighted via underline). In addition, the generated story from the NRETM model has more rational plots than the one from the T5 model. In the fourth story, the length of the first and fourth sentences are controlled to be 12 and 9. The outputs of NRETM model successfully obey these control constraints while the baseline model generates 11 and 13 tokens for the first and fourth sentences. In the last story, the second sentence generated by the NRETM model successfully generates six stop words (highlighted via underline). For this task, we are more concerned about the expression rate of predicate logic control constraints than the quality of the generated story. In addition to the case study, we have shown more quantitative analysis, and please refer to Sec. A.4 for details.

Table 9: Generated Example of the T5 and NRETM model in the Commonsense Generation task.

| **Task 2: Commonsense Generation** | |
|---|---|
| Input expression: | Copy(stone) & Copy(explain) & Copy(knife) & Copy(sharpen) |
| Logic expression: | Copy(stone) ∧ Copy(explain) ∧Copy(knife) ∧Copy(sharpen) |
| **T5**: | a man is sharpening a knife on a stone |
| **NRETM**: | a man explains how to sharpen a knife on a stone |
| Input expression: | Copy(stand) & Copy(map) & Copy(report) & Copy(front) & Copy(weather) |
| Logic expression: | Copy(stand) ∧ Copy(map) ∧ Copy(report) ∧ Copy(front) ∧ Copy(weather) |
| **T5**: | map showing where the weather is standing at the front |
| **NRETM**: | a man stands in front of a map reporting the weather |
| Input expression: | Copy(put) & Copy(lipstick) & Copy(talk) & Copy(lip) |
| Logic expression: | Copy(put) ∧ Copy(lipstick) ∧ Copy(talk) ∧ Copy(lip) |
| **T5**: | a woman puts lipstick on and talks about it |
| **NRETM**: | a woman is talking and putting lipstick on her lips |
| Input expression: | Copy(iron) & Copy(straighten) & Copy(demonstrate) & Copy(hair) |
| Logic expression: | Copy(iron) ∧ Copy(straighten) ∧ Copy(demonstrate) ∧ Copy(hair) |
| **T5**: | a woman straightens her hair with an iron and shows how to do it |
| **NRETM**: | a woman is demonstrating how to straighten her hair with an iron |
| Input expression: | Copy(bride) & Copy(stand) & Copy(bridesmaid) & Copy(groomsman) & Copy(groom) |
| Logic expression: | Copy(bride) ∧ Copy(stand) ∧ Copy(bridesmaid) ∧ Copy(groomsman) ∧ Copy(groom) |
| **T5**: | bride standing with her bridesmaids and groomsmen |
| **NRETM**: | the bridesmaids and groomsmen stand in front of the bride and groom |
| Input expression: | Copy(kitchen) & Copy(watermelon) & Copy(knife) & Copy(cut) |
| Logic expression: | Copy(kitchen) ∧ Copy(watermelon) ∧ Copy(knife) ∧ Copy(cut) |
| **T5**: | a knife cutting a watermelon in a kitchen |
| **NRETM**: | a man cutting a watermelon with a knife in the kitchen |

Table 9 shows selected examples from our T5 baseline and NRETM models in the Commonsense Generation task. Concepts that are missed in the baseline model outputs are in red. Words in blue are the key difference between the output of baseline and NRETM model. Note that we omit the synonyms for simplicity. Full Examples for this task can be found in Sec. A.3. Although the baseline model can correctly complete many Copy operations, it fails when the input combination is not commonly seen. For example, "explain" and "knife" in the first example. The baseline model also generates meaningless sentence when the inputs are complicated concepts combination in the second example. In addition, the baseline model cannot handle the case where some input concepts share the same prefix, such as "groom" and "groomsman" in the forth example. The baseline model seems to merge these morphological similar input concepts into a single concept and only mentions one of them in the outputs. Whereas the NRETM model successfully completes all of Copy operations.

Table 10 shows selected examples from our T5 baseline and NRETM models in document-level machine translation. In the first case, the mT5 baseline model produces duplicated sentences ("what happens when co2 emissions go up", in red). As a consequence, it fails to translate a few important chunks in the source sentences (see in Blue). This may due to the fact that the mT5 baseline model cannot handle long input documents well. While our NRETM model translates all source sentences into fluent English. Sentences in Green are missed by the baseline model but successfully translated by the NRETM model with the help of the predicate translateOnce. In the second case, the baseline model skips the important word "exchange" (see underline in the Input expression) in its translated text (highlighted in red). The NRETM model accurately translates this sentence (highlighted in blue). This shows that the NRETM model is more focused on the current sentence than the T5 baseline model.

Table 10: Generated Example of the mT5 and NRETM model for document-level machine translation.

| Task 3: Document-level Machine Translation |
| --- |

| | |
| --- | --- |
| Input expression: | 当前二氧化碳巨大的排放量将导致温度上升，温度的升高将引起一系列非常严重的后果，比如对天气的直接影响，或对生态系统的间接影响，生态系统无法应对剧烈变化的结果就是生态系统的全面崩溃。二氧化碳排放增加和温度升高究竟成怎样的关系，两者间的正反馈效应为何？这中间有一些不确定因素，但不多。至于全球变暖的具体负面影响有多严重，这无法完全确定，但肯定极其严重。translateOnce(1) & translateOnce(2) & translateOnce(3) |
| **mT5**: | what happens when co2 emissions go up – what happens when co2 emissions go up – is that the system collapses. what's the positive feedback between co2 emissions and temperature rising, and what's the negative feedback between co2 emissions and temperature rising?there's some uncertainty about how severe it is, but it's certainly severe. |
| **NRETM**: | what's going to happen when co2 emissions go up will cause temperatures to rise with a range of very severe consequences, such that, either directly or indirectly, ecosystems will collapse. there's some uncertainty about how this relationship, the positive feedback, between co2 emissions and temperature rise will work. how severe it will be, it's not entirely clear. |
| Input expression: | 我是跟着50个去往贾拉拉巴德战役的反叛者乘着卡车一起来的，当时我19岁，是一位住在佛罗里达州杰克逊维尔的素食主义者兼冲浪爱好者。我用我的黑色匡威的低邦鞋换了一双棕色的皮拖鞋，并且对着我看不太清的政府的坦克发了一枚火箭。这是我第一次到阿富汗。translateOnce(1) & translateOnce(2) & translateOnce(3) |
| **mT5**: | I was riding a truck with 50 rebels who had gone to Jalabard to fight the Taliban, and I was 19 years old, a vegetarian and surfer in Jacksonville, Florida. I took my black shoe boots and a brown leather shoe, and I threw a rocket at a government tank that I couldn't see. This was my first trip to Afghanistan. |
| **NRETM**: | I was accompanied by 50 counter-insurgents on a truck heading to Jamalabad, Florida, when I was 19 years old, a vegetarian and surfers living in Jacksonville, Florida. I exchange brown leather sandals with my black jackets, and I fired a rocket at the tanks of the government that I didn't see. This was my first time in Afghanistan |

## 5.2 Definition of Predicates

Figure 3 to 8 show the six kinds of predicates, including InSen, Order, Copy, TranslatedOnce, Len and StopWordCount, used in our framework. The code in figures is the pseudo code of the logical operator (i.e., executable programs), in which y refers to $\mathbf{y}_{:t}$, y_i is $y^i$, state is the default state status $\hat{q}_{it}^k$ and i_val is the additional intermediate values $\bar{q}_{it}^k$ . In our experiments, all of these logical operators are implemented using the Python programming language, and their source codes are not directly visible to the neural text generators. They only communicate with the neural text generators using the state flags. All predicates have State S1, indicating unfinished status and State S3, indicating finished status. As discussed in Sec 2.4, State S2 is an optional predicate-specific state. We will introduce the definition and role of State S2 for each of the above predicate if it exists in the captions.

| Predicate | Description |
|---|---|
| **InSen**$(x_i, y^k)$ | **If a phrase $x_i$ exists in the $k^{th}$ sentence in $y$** |

```python
def InSen(x, y_i, y):
    t = len(y)
    s = y.sen_count()
    if s == i and x not in y:
        state = S2
    elif s == i and x in y:
        state = S3
    else:
        state = S1
    return state
```

Figure 3: The definition of predicate InSen. The State S2 starts when the text generators start to generate $k^{th}$ sentence. This informs the model that it is possible to mention $x_i$ in the outputs.

| Predicate | Description |
|---|---|
| **Order**$(x_i, x_j)$ | **If phrase $x_i$ is before $x_j$ in the decoded sequence $y$** |

```python
def Order(x_a, x_b, y):
    x_a_s = y.IndexOf(x_a)
    x_b_s = y.IndexOf(x_b)

    if x_a in y and x_b not in y:
        state = S2
    elif x_a in y and x_b not in y and x_a_s < x_b_s:
        state = S3
    else:
        state = S1
    return state
```

Figure 4: The definition of predicate Order. The State S2 starts when the previous element $x\_a$ has already been mentioned in the outputs. This informs the model to mention $x\_b$ next.

| Predicate | Description |
|---|---|
| **Copy**$(x_i)$ | **If the decoded sequence $y$ contains phrase $x_i$** |

```python
def Copy(x, y):
    if x in y:
        state = S3
    else:
        state = S1
    return state
```

Figure 5: The definition of predicate Copy. There is no State S2 in the definition of Copy because there is no "partial copy" status.

| Predicate | Description |
|---|---|
| **TranslatedOnce(*x^i*)** | **If the *i^{th}* sentence in the source be translated only once** |

```python
def TranslatedOnce(x_i, y):
    s = y.sen_count()
    if s > i:
        state = S3
    elif s = i:
        state = S2
    else:
        state = S1
    return state
```

Figure 6: The definition of predicate $\mathrm{TranslatedOnce}$. The State S2 starts when $i^{th}$ sentence is being translated. This informs the model should pay attention to which source sentence.

| Predicate | Description |
|---|---|
| **Len(*y^i*,*l*)** | **If the length of *i^{th}* sentence in *y* is 1** |

```python
def Len(y_i, l, y):
    t = len(y)
    s = y.sen_count()
    s_pos = y_i.start_position()
    if s = i:
        state =S2;  i_val = l - (t-s_pos)
    elif s > i and Len(y_i) = l:
        state =S3;  i_val = 0
    elif s > i and Len(y_i) != l:
        state =S1;  i_val = (l-Len(y_i))
    elif s < i:
        state =S1;  i_val = l
    return (state, i_val)
```

Figure 7: The definition of predicate $\mathrm{Len}$. The State S2 starts when the text generator starts to generate $i^{th}$ sentence. We also explicitly inform the model of how many tokens are remaining for the current sentence. So they have State S2 with additional information $i\_val$.

| Predicate | Description |
|---|---|
| **StopWordCount(*y^i*,*s*)** | **If the count of stop words in *y^i* equaltos** |

```python
def StopWordCount(y_i, s, y):
    t = len(y)
    s_ = y.sen_count()
    s_pos = y_i.start_position()
    if s_ = i:
        state =S2; i_val = s - (stop_count(y{s_pos:t}))
    elif s_ > i and stop_count(y_i) = s:
        state =S3; i_val = 0
    elif s_ > i and stop_count(y_i) != s:
        state =S1; i_val = (l-stop_count(y_i))
    elif s_ < i:
        state =S1; i_val = s
    return (state, i_val)
```

Figure 8: The definition of predicate $\mathrm{StopWordCount}$. The State S2 starts when the text generator starts to generate $i^{th}$ sentence. We also explicitly inform the model of how many stop words are remaining for the current sentence. So they have State S2 with additional information $i\_val$.

### 5.3 Implementation Details For Each Evaluation Task

In this section, we will introduce the implementation details of all our evaluation tasks. In our experiments, we use three different pre-trained language model, T5-base, T5-Large and MBart-Large. We use the implementation of *huggingface transformers* [4]. We modify their decoder models to integrate our state matrix and use their provided model weights in our experiment. We only additional introduce the *State Matrix Encoder*. It is a one-layer transformer encoder. Its hidden size equals to the dimension of each head in the pre-trained transformer-based langauge models. The size of its FFN layer is 256. The number of its heads is 4.

**Controllable ROCStories Generation**    We first use RAKE algorithm (implemented by `https://github.com/csurfer/rake-nltk`) to extract storyline (i.e., key words and phrases) from the ground-truth stories. In the ROCStories dataset, each story has 5 sentences. For extracted storylines, we can easily find their original sentence index and ordering. We can also extract total length and stop word counts from each sentence in the ground-truth stories. We use these information to construct the training rules. For rules with only logic $\wedge$, we simply use these extracted ground-truth information as the predicate logic constraint. For rules with logic $\vee$, we create all cases with equal proportion in the training data. For example, for clause $\text{Len}(y^i, l_i) \vee \text{StopWordCount}(y^i, s_i)$, we create 33% of the training data only satisfy $\text{Len}(y^i, l_i)$, 33% of the training only satisfy $\text{StopWordCount}(y^i, s_i)$ and the remaining training data satisfy both of them. We can assign fake value for $l_i$ or $s_i$ for the above data argumentation. To improve the generalization of our pre-trained model, we freeze the parameters in the Self-Attention module and Feed-Forward Layers in each layer of the T5 decoder. This parameters freezing technology is applied to both T5 baseline models and the `NRETM` models in all of our experiments. We use constant learning rate $5e^{-5}$ and batch size 32 for this experiment.

**Commonsense Generation**    In the Commonsense Generation task, we first use NLTK toolkit to expand each input concept with all of its possible inflected forms, including plurals and different tenses. We further search the mention position of each input concept, including all of its inflected forms, on its corresponding ground-truth references. We use the same model and training setup in the Controllable ROCStories Generation task. We use constant learning rate $5e^{-5}$ and batch size 48 for this experiment.

**Document-level Machine Translation**    In the document-level Machine Translation, we split each documents into 2 - 4 trucks. Following the fine-tune setup in the original MBart paper, we use learning rate $3e^{-5}$. But we use batch size 8 and total training step 80k for our experiment.

---

[4] `https://github.com/huggingface/transformers`