# OpenReview forum: "Neural Rule-Execution Tracking Machine For Transformer-Based Text Generation"
_NeurIPS.cc/2021/Conference — NeurIPS 2021 Poster_

### Official Review · Reviewer_iHME · 2021-07-05

**Rating:** 6
**Confidence:** 4

**Summary:**

The paper proposes a unified framework to incorporate prior knowledge into the text generation process via predicate logic constraints. The text generation model is a transformer and constraints are also encoded by a separate transformer. The encoded constraints then become part of the attention mechanism, therefore it can be considered as an approach that uses “soft” constraints.

Contributions
- A general framework of incorporating prior knowledge as (soft) constraints into the decoding process for text generation.


**Limitations And Societal Impact:**

The paper's broader impact section discusses general potential benefits and issues of text generation (from large language models). It could maybe be tailored a bit better by discussing what effect this proposed work would have on the potential benefits and issues.

**Main Review:**

After Authors' Response
The authors' response was very thorough and addressed all my raised concerns, therefore I have increased my score.

Strength
-	The idea makes sense on a high level and having an efficient manner of incorporating constraints into the decoding process for text generation would be very beneficial for many tasks and applications.

Weaknesses
-	Some important points about the method and the experiments are left unclear (see also questions below).
-	The writing could be improved (see also Typos & Additional Questions below)
-	Multiple runs and significance tests are missing. This makes it hard to judge the improvements (Table 2 & 3).

Most Important Questions
-	Line 156: What is q_ij^k here exactly? I thought q_ij was a state flag, such as “2” or “0”. But you tokenize it and encode it, so it sounds more like it is something like “Copy(snow)”? (If it is the latter, then what is the meaning of tokenizing and encoding something like “Len(9)”?)
-	192: What exactly is storyline and what do you need it for?
-	The baseline takes the predicate logic constraints as input: How does T6 know what to do with these inputs? Was the model trained on this but without the NRETM module? Can you give an example of what the input looks likes? How do these inputs  guide which sentences should be generated? Looking at the datsset, it feels like one would need at least the first 2 sentences or so to know how to continue. Maybe this information is now in your constraints but it would be important to understand what they look like and how they were created. Is there no other suitable baseline for this experiment?
-	What is the overhead of your method compared to standard decoding approaches? (you mention GBS can only be used with T5-Base, so your method is more efficient? That would be important to point out)
-	What happens if the decoding process cannot find a sequence that satisfies all constraint?
-	Document-level MT: How do you know at test time whether the system translates a particular sentence or not?
-	How many sentences are misaligned by Doc-mBART25? What are the s-BLEU and d-BLEU values on the subset that NRETM aligns correctly and Doc does not?
-	Why was NEUROLOGIC not used as a comparison baseline?
-	What is dynamic vs static strategy? In which experiment did you show that dynamic works better than static (from conclusion)?


Typos & Additional Questions
-	Line 40: you could mention here that the examples will be translated into logic forms in the next section.
-	Paragraph starting at line 53: Why did you choose these datasets? How will they help evaluate the proposed approach?
-	Line 75: a and b should be bold faced?
-	83: “that used” -> “that are used”
-	83: “details” -> “for details”
-	Paragraph at line 86: At this point, the state matrix is unclear. What are the initial values? How can the state matrix be used to understand if a constraint is satisfied or not?
-	98: “take[s]” & “generate[s]”
-	108: “be all” -> “all be”
-	Paragraph at line 101: What is dynamic vs static strategy?
-	Paragraph at line 109: The state flag explanation would greatly benefit from an example. Does q_i refer to whether a particular U_i is satisfied?
-	Eq 2: What is the meaning of N? Can it change depending on the definition of U_k? Does it mean this constraint is not relevant for x_i?
-	133: Figure 1 should be Figure 2
-	Figure 2: What exactly do the “&” rows track?
-	Figure 2: Is the state flag matrix equal to the state matrix? If not, how do you go from one to the other?
-	Line 146: What does the inf in the superscript signify?
-	177: What is the symbolic operator?
-	Paragraph at line 194: Without understanding what a storyline is, it is not clear what the constraints are. An example might be helpful here.
-	Line 204: what is the ROUGH-L metric? Do you mean ROUGE-L?
-	Line 223: How do you obtain the morphological inflections for the concepts?
-	237: @necessity [of] integrating”
-	3.3: How exactly is the document-level MT done? Is the entire input document the input to T5?
-	293: “because” typo
-	3.4 where/how exactly is the sentence index used?


**Time Spent Reviewing:**

3

---

> ### Author Response · Authors · 2021-08-10
> **Response to Reviewer iHME**
>
> We thank you for your efforts in helping us point out most of the typos and unclear points in our paper one by one. The issues you pointed out can really be helpful for us to improve the clarity of our submission. We will correct all typos and all unclear points that you mentioned in our new version.
>
> **Multiple Run for Table 2 and Table 3**
>
> Thanks for pointing this out. We re-run our models in Tables 2 and 3 for five times. The below table shows the average (std.) performance. We conduct t-student test for these results, and all p-values are less than 0.01. We will add this result to our new version.
>
> | Model     | Metrics  | Baseline | NRETM        |
> |:---------:|:--------:|:--------:|:------------:|
> | CommonSen | CIDEr    | 163.6    | 174.7 (0.79) |
> |           | SPICE    | 32.4     | 33.7 (0.09)  |
> | MT        | Sen-BLEU | 23.8     | 24.8 (0.08)  |
> |           | Doc-BLEU | 29.6     | 30.3 (0.05)  |
>
> **Q: Line 156: What is $q_{ij}^k$ here exactly?**
>
> Actually, $\hat{q}\_{ij}^k$ (defined in eq. 2) is the state flag for each predicate and its value could be N,0,1, or 2. As shown in Figure 2, we show a concrete example of our State Flag Matrix. Each cell in that matrix is a string corresponding to $q_{ij}$ and $q_{ij}^k$ is the $k^{th}$ tokens in $q_{ij}$. We do realize that they are very similar to each other. We will update the notation to avoid this misunderstanding.
>
> **Q: Controllable ROC Storyline Baseline Question The baseline takes the predicate logic constraints as input ...**
>
> **Q: Paragraph at line 194: Without understanding what a storyline is...**
>
> Yes. The baseline models in Table 1 take those constraints as input and do not have the NRETM module.
> The storyline is the keywords or keyphrases of the story. The controllable ROC story task in section 3.1 is to use these keywords or phrases as inputs (as well as predicate logic constraints) to generate the whole story. This setting largely follows previous work [1]. We start from the storyline and construct predicate logic constraints (i.e., InSen(walking around, 2) \& InSen(school yard, 2)). The models use these constraints to generate the whole story. Note that the goal of this experiment is not to show we can generate better stories but to show that our proposed NRETM module can help the T5 models to follow the given constraints. To the best of our knowledge, we are the first work to incorporate predicate logic constraints into pre-trained seq2seq text generators. So our provided baselines should be reasonable.
>
> **Q: Computational overhead**
>
> We compare the inference time (measured in minutes, M) for the following models on the test split of commonsense generation task. All of them use the beam search decoding algorithm with beam size 5. Adding NRETM components to T5-Base and T5-Large approximately double the inference time. While the Grid Beam Search (GBS) algorithm uses a much longer inference time. We will add this result to our new version.
>
> |                   | Inference Time (M) |
> |:-----------------:|:------------------:|
> | T5-Base-Baseline  | 1.05               |
> | T5-Base-GBS       | 84                 |
> | T5-Base-NRETM     | 2.27               |
> | T5-Large-Baseline | 1.32               |
> | T5-Large-NRETM    | 2.6                |
>
> **Q: Cannot satisfy all constraints**
>
> Our proposed NRETM model does not impose hard constraints on the text decoder but instead provides State Flag Matrix as input features describing rule execution intermediate values to the text decoder. That is, no explicit effect when the models fail to satisfy the constraints. It is possible that our text generators decide to stop the generation before completing all constraints. In our experiments, our proposed NRETM model has less than 1% chance not to complete all constraints.
>
> **Q: Document-level MT: translated particular Sentence**
>
> The benchmark (TED15 Zh-En) we used in this experiment provides sentence-to-sentence translation from Chinese to English. That is, for each document, the $i^{th}$ sentence in the generated output corresponds to the $i^{th}$ sentence in the source input documents. Based on this prior knowledge, we add an additional End-Of-Sentence token at the end of each sentence to the training data. Once our NRETM model finishes the $i^{th}$ sentence (generating an end-of-sentence token) in the decoder, we assume that the $i^{th}$ sentence in the encoder has been translated. The goal is to guide the translation model to focus on the correct parts of input documents.
>
> **Q: misaligned Sentence s-BLEU and d-BLEU**
>
> There are 93 sentences that are misaligned by Doc-mBART25. Note that this number is calculated based on whether the translated documents have the same number of sentences as the ground-truth documents. We also see cases where the number of sentences is correct, but the content is mismatched in the documents. As our proposed NRETM model successfully align all documents, we calculate s-BLEU and d-BLEU score over these 93 sentences. As shown in the below table, the proposed Doc-mBART25 + NRETM model outperforms the Doc-mBART25 baseline in the s-BLEU by a large margin (18.5 vs. 6.1), showing the importance of alignment from our NRETM model. We will add this result to our new version.
>
> | Model               | s-BLEU | d-BLEU |
> |:-------------------:|:------:|:------:|
> | Doc-mBART25         | 6.1    | 20.5   |
> | Doc-mBART25 + NRETM | 18.5   | 21.9   |
>
> **Q: NEUROLOGIC Baseline**
>
> As this is pointed out by several reviewers, we report the comparison result with NEUROLOGIC baseline in the Overall Response. Our proposed NRETM model outperforms this strong baseline. Thanks for pointing this important baseline out. This would significantly improve our paper.
>
> **Q: Paragraph at line 101: What is dynamic vs static strategy?**
>
> **Q: What is dynamic vs static strategy? ...**
>
> Our T5 baseline model can be viewed as a static strategy because it has seen all input constraints in the encoder and these input constraints are always static during text generation. We show that updating the intermediate execution result for these constraints in the decoder works better than the encoder-only baseline.
>
> **Q: Dataset Selection on line 53**
>
> These three tasks all involve constraints or prior knowledge, allowing us to verify the effectiveness of our proposed NRETM model. In the ROC story generation task, we have complicated predicate constraints over the story structure. In the commonsense generation task, mentioning all input concepts are the constraints. In document MT, the prior knowledge is that model should translate input sentences one by one.
>
> **Q: Paragraph at line 86**
>
> The initial value for state matrix can be  N (not relevant), 0 (not satisfied) or 1 (in progress). Once predicates finish, the corresponding value ends up with 2. This is described in eq. 2.
>
> **Q: Does $q_i$ refer to whether a particular $U_i$ is satisfied?**
>
> Yes. $q_{i,t}$ refers to the status of particular $U_i$.
>
> **Q: Eq 2: What is the meaning of N? ...**
>
> Yes. It means this constraint is not relevant for $x_i$. This entirely depends on the definition of $U_k$. For example, the length operator (i.e., Len(t)) is relevant to all tokens; while the copy operator (i.e., copy(x)) is only relevant to the encoder tokens that describe it.
>
> **Q: Figure 2: What exactly do the \& rows track?**
>
> This is a basic logic operator that connects two logic operators. It tells the model both constraints need to be satisfied. This can be omitted when all constraints are in the same format but is important when the input includes different forms of constraints (e.g., some constraints are connected by "and" and some constraints are connected by "or").
>
> **Q: Figure 2: Is the state flag matrix equal to the state matrix?**
>
> Yes. They are the same thing. We will unify the term in our new version.
>
> **Q: Line 146: What does the inf in the superscript signify?**
>
> Sorry, this is a mistake. We will remove this in our new version.
>
> **Q: Line 204: what is the ROUGH-L metric? Do you mean ROUGE-L?**
>
> Yes. You are right. We will correct this typos in our new version.
>
> **Q: 177: What is the symbolic operator?**
>
> Symbolic operator refers to the logic operator we defined in this paper. All of these logic operators are defined in a symbolic way (i.e., they are not trainable neural embeddings). Please refer to Sec 5.2 to see the definition in our supplementary materials.
>
> **Q: morphological inflections on Line 223**
>
> We collect morphological inflections for each concept from wordnet.
>
> **Q: Details for Document-level MT**
>
> Sec 5.5 in our supplementary material describes these in more detail.
>
> **Q: 3.4 where/how exactly is the sentence index used?**
>
> In 3.4, The sentence index are encoded in the encoder of T5 model in both baseline and NRETM model. The experiment is to train the model to put storyline (i.e., words or phrases) in the third, fourth, fifth sentence and test the model to put storyline in the first and second sentence. Our goal is to show that our NRETM model can execute rules that are not seen in the training data, showing the zero-shot ability of our proposed model.
>
>
> [1] Nanyun Peng, Marjan Ghazvininejad, Jonathan May, and Kevin Knight.   Towards controllable story generation. InProceedings of the First Workshop on Storytelling, pages 43–49, New Orleans, Louisiana, June 2018. Association for Computational Linguistics.

---

> ### Author Response · Authors · 2021-09-02
> **We are happy to have more discussions with you**
>
> We thank you for your efforts and constructive comments on our paper. We have tried to respond to each of those issues that you mentioned in the review. We are happy to have more discussions with you and provide more details to address your concerns. We would like to know if our current rebuttal has addressed your concern. In addition, please feel free to raise any additional questions/concerns over our work. We would provide you with additional responses and promise to include most of these clarifications in the final revised version to make the paper more clear.

---

> > ### Comment · Reviewer_iHME · 2021-09-13
> > **Thanks for the detailed answers**
> >
> > Thank you for answering my questions in such a detailed manner, I have increased my score.

---

> > > ### Author Response · Authors · 2021-09-13
> > > **Thank you!**
> > >
> > > Thank you for your generous response. We appreciate your effort in reviewing our submission. We are very grateful to see that our response resolve your concerns in the initial review processing. We do spend a significant amount of effort in preparing to respond to your questions. We later found that these questions are very helpful in improving the quality of our submission. We will incorporate our response into our final version.

---

### Official Review · Reviewer_ysKc · 2021-07-09

**Rating:** 6
**Confidence:** 3

**Summary:**

Interesting problem, but many issues along the way

**Limitations And Societal Impact:**

The applications of the proposed methods are limited. The paper states that "because in the training data at the position of EOS token in the ground-truth output, all elements in the state matrix must be set to the satisfied, NRETM can learn the rules by itself." However, if the training data contains some noise, can the proposed methods work as well? More analyses about this should be conducted.


**Main Review:**

This work aims to find a way to impose the symbolic rules into powerful neural networks. To this end, it proposes a unified and scalable approach, named NRETM, to perform fine-grained controllable text generation and incorporate multiple prior knowledge.

Strength:

1. This paper proposes a framework to unify the constrained generation and general text generation with prior knowledge incorporation.

2. Experimental results on story generation, machine translation, and commonsense generation partially demonstrate the effectiveness of the proposed method.

Drawbacks:

1. The novelty of this work is not clear. It seems to extract the intermediate values of the rules to the model, which is helpful for the model to learn the relationship between the rules and the final outputs. If my understanding is correct, is it not surprising that NRETM can make better results since additional prior knowledge can usually benefit the generation process.

2. The experiments lack rigor. The author should add some competitive baselines for comparison. This paper mainly compares NRETM with pre-trained model T5. But T5 can not utilize prior knowledge efficiently. Feeding the predicate logic constraints as inputs directly is not a suitable way to impose prior knowledge.



The manuscript has some typos:
1) eq. 1 P(y|x) is not defined.
2) line 103, in 1 -> in Figure 1.
3) line 116, consist -> consists.

**Time Spent Reviewing:**

4 hours

---

> ### Author Response · Authors · 2021-08-10
> **Response to Reviewer ysKc**
>
> We appreciate your efforts in the review process. We are also glad for the acknowledgment that the problem we are working on is interesting. We also thank you for pointing typos in our paper which are helpful for us to improve the paper quality. We will fix all of these typos in our new version.
>
> **Core Contribution**
>
> We agree with your understanding of the core contribution of our paper in terms of methodology. As we mentioned in the Overall Response (i.e., **Core Contribution and Novelty**), it is also important to note that another part of our core contribution is that we are the first paper to provide a general solution to train transformer-based seq2seq text generators to follow predicate logic constraints. In addition, as shown in the experiment in the Overall Response (i.e., **Encoder Update Baseline**), it is very important to incorporate the rule execution intermediate values in the text decoder as we did in our paper. Updating this information in the text encoder does not help the models to complete constraints.
>
> **Baseline Setup**
>
> You also mentioned our baseline setup. As discussed in the Overall Response, we add NEUROLOGIC and the encoder update baseline and compare them with our proposed model. We hope this reduces some of your concerns on the baseline issue. In addition, there is no previous work that can use these complicated predicate logic constraints as input. Feeding these constraints directly to the T5 model is a reasonable baseline. The raw T5 models indeed do not understand the predicate logic constraints. However, in our experiment, the baseline T5 model is fine-tuned with these constraints as input. This baseline model indeed learns to use these constraints to guide the output to some extends. As shown in Table 8 (Section 5.4 in our supplementary materials), although this baseline always performs worse than our proposed NRETM model, its CSR reaches over 75\% when we relax the success condition to $\pm$ 1 and $\pm$ 2. In summary, this baseline model cannot learn to control the output text accurately, but it learns to control the text in a fuzzy way.
>
>
> **Training Data Noise**
>
> We conduct the following training data noise experiments in the commonsense generation task (Sec 3.2): we corrupt the training data by replacing the input commonsense keywords with a random sampled one under the probability 5%, 10%, 15%, 25%, and 50%. (Dev and Test Split remain unchanged.) In all noise level, our proposed NRETM model still successfully mention more concepts and achieve higher CIDEr and SPICE score than the baseline model in the same noise level. More importantly, the model with 25% noise level can still outperform the ordinary baseline (i.e., the T5 model in the 0% noise level). This shows that our NRETM model is relatively robust to the training data noise. Finally, the main goal of NRETM model is to incorporate constraints that are satisfied by training data into transformer-based seq2seq text generators. So it is reasonable to assume that in practice, the noise level should be relatively low (e.g., 0% - 10%). We will add this result to our new version.
>
> | Noise Level | Model | CIDEr | SPICE | Constraint (All) |
> |:-----------:|:-----:|:-----:|:-----:|:----------:|
> | 0.0         | T5    | 163.6 | 32.4  | 93.9       |
> |             | NRETM | 175.0 | 33.8  | 99.3       |
> | 0.05        | T5    | 162.4 | 31.8  | 92.9       |
> |             | NRETM | 169.8 | 33.1  | 97.4       |
> | 0.1         | T5    | 163.0 | 32.1  | 92.3       |
> |             | NRETM | 167.6 | 33.1  | 95.0       |
> | 0.15        | T5    | 162.0 | 32.3  | 93.1       |
> |             | NRETM | 170.1 | 32.9  | 96.5       |
> | 0.25        | T5    | 161.2 | 32.0  | 91.2       |
> |             | NRETM | 166.4 | 32.5  | 94.7       |
> | 0.5         | T5    | 144.2 | 29.4  | 84.4       |
> |             | NRETM | 145.9 | 29.5  | 87.5       |

---

> > ### Comment · Reviewer_ysKc · 2021-08-27
> > **Thanks for your response**
> >
> > The author's response has addressed part of my concerns. I have improved my score.

---

> > > ### Author Response · Authors · 2021-08-27
> > > **Thank you**
> > >
> > > Once again, we appreciate your efforts in the review process, in particular for further responding to our updated results in the rebuttal.

---

### Official Review · Reviewer_HmFk · 2021-07-16

**Rating:** 6
**Confidence:** 4

**Summary:**

The authors propose an approach aimed at encouraging transformer-based sequence-to-sequence models to respect constraints on the text they generate. In particular, the authors propose a particular syntactic representation of whether propositional logic-style constraints have been satisfied in some (partially) generated text, which is tracked throughout decoding. This syntactic representation is a function both of the current decoding step as well as the index of the particular constraint, and is updated dynamically as decoding progresses. The authors tokenize and encode these representations, and use them inside a relative-position-style attention mechanism. The authors fine-tune their models using these dynamic representations and they show that this allows for better control over various aspects of the generated text on RocStories, Commongen, and Zh-En document level translation tasks, with comparable or better quality.

**Limitations And Societal Impact:**

Yes

**Main Review:**

This paper tackles an interesting problem, and it obtains good results. Another nice thing about the paper is the authors largely consider constraints that are *not* easy to implement with just a simple constrained beam search. In particular, rather than constraints which disallow certain tokens or alignments, the authors consider constraints that require the presence or a particular number of certain words, which are much more challenging to guarantee with standard decoding algorithms.

In terms of contribution, the core contribution of the paper appears to be the proposal of an approach to dynamically and explicitly updating the decoder in response to whether or how much the text generated so far conforms with some pre-specified constraints.  While this sort of dynamic updating seems quite reasonable, similar ideas have been proposed before (as the authors note), such as in the case of coverage attention (Tu et al., 2016). Accordingly, I think the proposed approach would feel more compelling if the authors could argue that it is superior to other approaches to updating the decoder in response to how well constraints have been satisfied in the generated text so far. However, I don't think there are any such comparisons. In particular, it seems the main baselines the authors consider are either a baseline seq2seq model with the constraints encoded statically on the source side (Table 1) or a baseline seq2seq model with a constrained decoding algorithm (Table 2). As such, it's hard to tell how crucial the various modeling/encoding choices the authors have made in implementing their approach are. (A natural baseline not considered, for instance, might involve dynamically updating the constraint tokens consumed by the encoder in the baseline in Table 1 instead of using the authors' proposed attention model).

In terms of presentation, I think the paper is largely clear, although I would encourage the authors to emphasize earlier on that their method involves fine-tuning the model rather than simply implementing constraints at decoding time; I think the fact that fine-tuning is involved only really becomes clear on the bottom of page 5.

Minor:
Are the script 'U's on like 116 different from the non-script upper case 'U's on line 109?

Update after response from authors: thanks for your response; I'm increasing my score in view of the new baseline results.

**Time Spent Reviewing:**

5

---

> ### Author Response · Authors · 2021-08-10
> **Response to Reviewer HmFk**
>
> We appreciate your efforts in the review process. We are also glad you think our paper tackles an interesting problem and obtains good results.
>
> **Mentioning Fine-tuning earlier**
>
> We agree with you that fine-tuning is an important part of our approach. With fine-tuning, our proposed model can work well with various logic operators. In addition, fine-tuning also make a difference between our model and the NEUROLOGIC baseline. Please note that we do compare our proposed NRETM model with this inference-only NEUROLOGIC baseline. The experiments show that our proposed NRETM model outperforms this NEUROLOGIC baseline. Details please refer to the Overall Response (i.e., **NEUROLOGIC Baseline**). This is a good suggestion and we will update our draft to emphasize the usage of fine-tuning in the proposed NRETM model.
>
> **Encoder Update Baseline**
>
> *We thank you for pointing this baseline*. Given that we are the first work to incorporate predicate logic constraints, we believe this is an important baseline to compare with. We believe this is a very important suggestion to improve our paper.  We also note that some other reviewers also have similar concerns over our baseline setup. **To resolve this common concern, we move these added experiments as well as analysis to Overall Response.** The experiment results in two benchmarks (see below tables) show that this baseline model cannot follow the constraints as well as our proposed NRETM model.  This shows that simply extracting rule execution intermediate values is not enough. We hypothesis that this is because the encoders cannot effectively broadcast rule execution intermediate values to the text decoders. In addition, our relative-position style method allows these intermediate values to be well aligned with the corresponding predicates in the encoder, allowing accurate control over text generation output.  Please refer to the Overall Response about the Encoder Update Baseline result for More details. We will add this result to our new version.
>
> | Predicate Logic Constraint |      Model      |  R-L |  CSR |
> |:--------------------------:|:---------------:|:----:|:----:|
> |             (1)            |   T5-Baseline   | 33.1 | 98.7 |
> |                            |  T5-Update-Enc  | 33.0 | 98.6 |
> |                            | T5-NERTM (Ours) | 33.3 | 98.8 |
> |             (2)            |   T5-Baseline   | 49.6 | 97.0 |
> |                            |  T5-Update-Enc  | 49.6 | 96.9 |
> |                            | T5-NERTM (Ours) | 49.5 | 97.4 |
> |             (3)            |   T5-Baseline   | 32.9 | 18.5 |
> |                            |  T5-Update-Enc  | 32.9 | 16.5 |
> |                            | T5-NERTM (Ours) | 32.6 | 84.5 |
> |             (4)            |   T5-Baseline   | 33.0 | 23.5 |
> |                            |  T5-Update-Enc  | 33.1 | 21.2 |
> |                            | T5-NERTM (Ours) | 33.1 | 49.4 |
> |             (5)            |   T5-Baseline   | 32.9 | 11.0 |
> |                            |  T5-Update-Enc  | 33.2 |  7.9 |
> |                            | T5-NERTM (Ours) | 33.0 | 35.9 |
>
> |       Model      | CIDEr | SPICE | Constraint (All) |
> |:----------------:|:-----:|:-----:|:----------:|
> | T5-Base-Baseline | 159.5 |  31.9 |    94.0    |
> |   T5-Update-Enc  | 163.2 |  32.1 |    95.1    |
> |       Ours       | 169.8 |  32.7 |    99.3    |
>
> **Q: Are the script 'U's on like 116 different from the non-script upper case 'U's on line 109?**
>
> They are different. The one on line 116 represents unique predicates while the one on line 109 represents an element in the predicate list. We do this to emphasize that users can use a predicate for multiple times in their predicate logic constraints. The same type of predicates uses the same logical operators.

---

> > ### Comment · Reviewer_HmFk · 2021-08-27
> > **Thanks for your response**
> >
> > I've updated my score in view of the new baseline numbers.

---

> > > ### Author Response · Authors · 2021-08-27
> > > **Thank you**
> > >
> > > We thank your insightful review, including pointing out this important baseline, for our submission. We are also happy to see your acknowledgment of our new results in the rebuttal.

---

### Official Review · Reviewer_EVbC · 2021-07-16

**Rating:** 7
**Confidence:** 3

**Summary:**

This paper proposes a general framework, NRETM, for the conditional transformer-based seq2seq model. The paper first uses a state flag to indicate the expression progress of each predicate in the constraints set as a concatenation of logic trackers. The paper uses a one-layer transformer encoder to encode the state flag.  The paper utilizes the state matrix as a relative position to integrate it into the model. The results in three experiments show that the NRETM improves the transformer performance.

**Ethics Review Area:**

["Inappropriate Potential Applications & Impact  (e.g., human rights concerns)"]

**Limitations And Societal Impact:**

The paper describes the limitation and social impacts in the last section. Some limitations are addressed. The potential misuse of this framework is also stated.

**Main Review:**

Strengths:
1. The paper focus on an interesting problem about constraint-based generation tasks. The paper solves this problem by proposing a new Neural Rule-execution tracking machine that tries to incorporate rules into current transformer-based seq2seq generation models. The paper utilizes the logic tracker as relative position information to guide the model in the generation The experiment results over three different tasks seem to be promising.

2. The paper lists detailed experiment settings for three different tasks. The paper also discusses the ability of the model in a zero-shot setting. The paper includes code as a supplement and put more generation results as case studies for three different tasks. Those examples show that the model captures those constraints through the proposed framework,

Weaknesses:
1. Section 2.4 is a little bit confusing. Even though the paper provides Figure 2 as a running example of the NRETM model with three logic constraints, it still takes me some time to understand all the concepts introduced in this section. It would be better to reorganize the content. For example, a logic tracker should be introduced before the state flag which can help readers to better understand the concepts.

2. In section 3, the paper only compares their model to a limited baseline. The paper needs to add some baseline mentioned in related work such as NEUROLOGIC DECODING (Liu et al., 2020). In section 3.2, the constraint satisfaction metric is not clearly defined. The evaluation metrics for all/novel/seen constraints are also absent. The evaluation metrics are also limited. For story generation, rouge L is not able to cover all the details. Similarly, CIDEr can also show some limited aspect of generation results. It would be better to include more automatic metrics to show a more comprehensive view of the generation quality. such as Bertscore(Zhang et al., 2019), BLEU, etc.

3. The generalizability of the framework is unclear. It seems that the model can only be applied with several pre-defined logic operators. Those logic operators

Lu, X., West, P., Zellers, R., Bras, R. L., Bhagavatula, C., & Choi, Y. (2020). Neurologic decoding:(un) supervised neural text generation with predicate logic constraints. arXiv preprint arXiv:2010.12884.
Zhang, T., Kishore, V., Wu, F., Weinberger, K. Q., & Artzi, Y. (2019). Bertscore: Evaluating text generation with bert. arXiv preprint arXiv:1904.09675.

**Needs Ethics Review:**

Yes

**Time Spent Reviewing:**

5

---

> ### Author Response · Authors · 2021-08-10
> **Response to Reviewer EVbC**
>
> We thank you for the review and are grateful for the time you spent with our submission. We are also glad for the acknowledgment that the problem we are working on is interesting.
>
> **Update Sec 2.4**
>
> We agree that this section has introduced a lot of concepts, which can be simplified in some ways. We also think introducing Logic Tracker first makes more sense than our current version because each State  Flag combines the output of a few Logic Trackers. We will update Sec 2.4 based on your suggestions in our next version.
>
> **NEUROLOGIC baseline**
>
> Following your suggestion, we add NEUROLOGIC as a new baseline. Please refer to NEUROLOGIC Baseline in the Overall Response for more details. Besides, we note that you are interested in BERT Score and BLEU of our models. So we additional report BERT Score and BLEU for the NEUROLOGIC baseline in the below table. The NEUROLOGIC baseline outperforms our T5 baseline but achieves a lower score than our proposed NRETM model.
>
> |         Model         | BERTScore | BLEU-1 | BLEU-4 | CIDEr | SPICE | Constraint (All) |
> |:---------------------:|-----------|--------|--------|:-----:|:-----:|:----------:|
> |        T5-Large       | 94.6      | 72.3   | 30.6   | 163.6 |  32.4 |    93.9    |
> | T5-Large + NEUROLOGIC | 94.6      | 73.4   | 31.3   | 169.7 |  32.3 |    99.0    |
> |    T5-Large + NRETM   | 94.8      | 74.9   | 32.2   | 175.0 |  33.8 |    99.4    |
>
> **Constraint Type in Sec 3.2**
>
> In Section 3.2, Table 2, each constraint corresponds to an input concept. Some of these input concepts are seen in the training data; while others are not encountered in the training data. We classify the constraints in test split as seen constraints if the corresponding input constraints are seen in the training data; as novel constraints if the corresponding input constraints are not seen in the training data. All constraints refer to all of the input concepts in the test split.
>
> **BERTScore and BLEU result**
>
> We additionally report BERTScore/BLEU for the models in Table 1 and Table 2, as shown below. These newly added metrics report a similar trend reflected by the results of our submitted experiments: in the commonsense generation task, adding NRETM to T5-Base and T5-Large improves both generated text quality and constraint successful ratio. In the controlled ROC story generation task, adding NRETM improves CSR by a large margin without degrading generated story quality.
>
> | Predicate Logic Constraint | Model | BERT Score | BLEU-1 | BLEU-4 | ROUGE-L |  CSR |
> |:----------:|:-----:|:----------:|:------:|:------:|:-------:|:----:|
> |     (1)    |   T5  |    86.7    |  29.7  |   9.3  |   33.1  | 98.7 |
> |            | NRETM |    86.8    |  29.5  |   9.3  |   33.3  | 98.8 |
> |     (2)    |   T5  |    89.4    |  44.5  |  20.3  |   49.6  | 97.0 |
> |            | NRETM |    89.3    |  43.6  |  19.6  |   49.5  | 97.4 |
> |     (3)    |   T5  |    86.7    |  33.4  |  10.3  |   32.9  | 18.5 |
> |            | NRETM |    86.6    |  33.0  |  10.0  |   32.6  | 84.5 |
> |     (4)    |   T5  |    86.9    |  32.7  |  10.2  |   33.0  | 23.5 |
> |            | NRETM |    86.8    |  33.2  |  10.3  |   33.1  | 50.9 |
> |     (5)    |   T5  |    86.8    |  33.5  |  10.4  |   32.9  | 11.0 |
> |            | NRETM |    86.8    |  33.9  |  10.5  |   33.0  | 35.9 |
>
> |       Model       | BERT Score | BLEU-1 | BLEU-4 | CIDEr | SPICE | Constraint (All) |
> |:-----------------:|:----------:|:------:|:------:|:-----:|:-----:|:----------:|
> | T5-Base-Baseline  | 94.5       | 71.2   | 28.9   | 159.5 | 31.9  | 94.0       |
> | T5-Base-NRETM     | 94.5       | 73.9   | 30.1   | 169.8 | 32.7  | 99.3       |
> | T5-Large-Baseline | 94.6       | 72.3   | 30.6   | 163.6 | 32.4  | 93.9       |
> | T5-Large-NRETM    | 94.8       | 74.9   | 32.2   | 175.0 | 33.8  | 99.4       |
>
> **Generalization Issue**
>
> As mentioned in the paper, our proposed NRETM models can work with various predicate logic constraints. It is straightforward to construct new constraints to fulfill user requirements using the basic logic operator (i.e., "and", "or" and "not") and the existing 6 pre-defined logic operators described in our paper. In addition, our proposed NRETM models can work well with various self-defined logic operators. Users can easily expand the logic operators in NRETM by 1) writing the executable programs (e.g., in Python or Java) for their logic operators in the Logic Tracker. Please note that we have provided the pseudo-code for these logic operators in our supplementary materials. Our submitted source code also includes the corresponding implementations (in prior\_tasks.py); 2) fine-tuning the pre-trained language models with these new logic operators.

---

> > ### Comment · Reviewer_EVbC · 2021-09-01
> > **Thanks for your response**
> >
> > Thank you very much for your response. The author's response has addressed my concern by adding new baselines and additional evaluation metrics. I have updated my scores.

---

> > > ### Author Response · Authors · 2021-09-02
> > > **Thank you for your support**
> > >
> > > We thank you for your effort and support for our submission in this review process. Your comments are helpful in improving the quality of our work.

---

### Review · Ethics_Reviewer_x9DD · 2021-08-10

**Recommendation:**

The authors should simply acknowledge the complexity of risk mitigation for such systems with a statement very similar to the one I used in the body of the ethical review:   Mitigating the risks of these systems is an extremely complex socio-technical problem that many are working to understand and solve.

**Ethical Issues:**

Yes

**Ethics Review:**

The Ethical risks associated with transformer-based text generation are widely discussed in multiple high-quality papers. [1][2]. It is good that the authors acknowledge this in their broader impact section.

However, the Ethical weakness of this paper is that it has the potential to mislead uninformed readers  by grossly understating the complexity of mitigating the risks of neural conversational systems. It does so with the single line the Broader Impact section includes on risk mitigation in such systems:  "In order to mitigate these risks, AI systems could be leveraged to fight against misleading content and harassing material."

Mitigating the risks of these systems is an extremely complex socio-technical problem that many are working to understand and solve.  The fact that this is not acknowledged is a key ethical weakness.

[1] On the Dangers of Stochastic Parrots: Can Language Models Be Too Big? -  https://dl.acm.org/doi/pdf/10.1145/3442188.3445922
[2]  The social impact of natural language processing. In Proceedings
of the 54th Annual Meeting of the Association for Computational Linguistics (Volume 2: Short
Papers), pp. 591–598, 2016.

---

> ### Author Response · Authors · 2021-08-18
> **response to Ethics Reviewer x9DD**
>
> Thanks for your insightful Ethics Review.
>
> We agree that the Statement "In order to mitigate these risks, AI systems could be leveraged to fight against misleading content and harassing material." is too simple. Mitigating the risks of our proposed systems could be extremely complicated. We update the above sentence into "In order to mitigate these risks, it is possible to use AI systems to fight against misleading content and harassing material. However, as discussed in previous work [1,2], mitigating these risks could be an extremely complex socio-technical problem that many are working to understand and solve."
>
> [1] On the Dangers of Stochastic Parrots: Can Language Models Be Too Big? - https://dl.acm.org/doi/pdf/10.1145/3442188.3445922
> [2] The social impact of natural language processing. In Proceedings of the 54th Annual Meeting of the Association for Computational Linguistics (Volume 2: Short Papers), pp. 591–598, 2016.

---

> > ### Comment · Ethics_Reviewer_x9DD · 2021-08-25
> > **Thank you to authors for incorporating recommendations..**
> >
> > Thank you for your consideration of my comments and agreeing to incorporate the recommendations made.

---

### Review · Ethics_Reviewer_uMGq · 2021-08-13

**Recommendation:**

I believe these matters can be addressed in the current version of the paper. In particular, I urge the authors to avoid general optimistic speculations such as " it could be used in a broader range of more complex scenes (e.g., financial report generation) due to its advantages of combining multiple rules to generate high-quality texts. On the positive side, more efficient text generation can make these technologies more available to the general public. Our model verified on machine translation can benefit commercial and humanitarian translation services to help overcome language barriers." Statements like these run the risk of encouraging off-the-shelf usages of the method in high-stakes situations that can impact people's livelihood. Instead, the authors must acknowledge that while their method shows promise on several limited benchmarks, deployment in the real world requires a careful analysis of potential societal benefits and harms (e.g., the harms associated with furthering negative stereotypes against certain vulnerable groups).

**Ethical Issues:**

Yes

**Ethics Review:**

The first reviewer raised concerns about inappropriate potential applications of the proposed method. My understanding is that they are referring to some of the potential use-cases authors mention in their paper, including translation. Authors must acknowledge the limitations and potential harms their tool can lead to---impacting especially vulnerable individuals and communities.

---

> ### Author Response · Authors · 2021-08-18
> **response to Ethics Reviewer uMGq**
>
> Thanks for your careful Ethics Review and we believe these reviews are helpful for our submission to avoid ethical issues.
>
> We agree that the Statement "it could be used in a broader ......translation services to help overcome language barriers." could be too specific and encourage the direct application of our proposed method in harmful situations. We will update the above text into "while our proposed method achieves promise performance on several benchmarks, deployment of our method in the real world requires a careful analysis of potential societal benefits and harms (e.g., the harms associated with furthering negative stereotypes against certain vulnerable groups)"

---

### Author Response · Authors · 2021-08-10
**Overall Response**

Dear Reviewers, we greatly appreciate your efforts for careful reading, particularly for your thoughtful and insightful comments on our paper. We promise to revise the paper to clear the issues you mentioned in the comments in our new version.

**Core Contribution and Novelty**

We note that some of the reviewers have concerns over the novelty of our paper. The core contribution of our paper is two-fold:

1) to the best of our knowledge, we are the first to propose a general framework that incorporates control signal and prior knowledge, formulated as predicate logic constraints, into transformer-based seq2seq text generation models;

2) we train (or fine-tune) the transformer-based seq2seq text generation models to follow the predicate logic constraints(i.e., control signal or prior knowledge) by dynamically updating the rule execution intermediate progress value to the text decoder.

To the best of our knowledge, NEUROLOGIC [1] is the most relevant work that also uses predicate logic constraints to control the text generators. However, we are different with NEUROLOGIC in:

1) NEUROLOGIC only provides control constraints over the text generators. Instead, our proposed NRETM model is a general framework that provides control constraints (e.g., copy or not copy words) and prior knowledge (e.g., translating sentences one by one). NEUROLOGIC can be viewed as a special case of our NRETM model.

2) NEUROLOGIC is an inference-only algorithm that only controls the model to generate or avoid specific words or phrases at decoding time; while our proposed NRETM model fine-tunes the pre-trained transformer-based seq2seq text generators with the predicate logic constraints;

3) NEUROLOGIC only supports the "copy" logic operator (i.e.,  to generate or not to generate specific words or phrases), while our NRETM model is a general framework that supports various logic operators. We list 6 kinds of logic operators in this paper, and it is also possible for users to expand new logic operators.

**NEUROLOGIC Baseline**

As suggested by reviewers, we add the comparison with the NEUROLOGIC approach. NEUROLOGIC can be viewed as a special case of our NRETM model that only supports the ``copy'' logic operator. However, in the ROC story generation task, we need to control the length of the generated text and copy words in specific sentences; in the document NMT, we need translation once logic operator. So this important baseline is only comparable in the commonsense generation task. The comparison results can be found in the below table. We apply NEUROLOGIC to the same T5 model that we used in this paper. Both NEUROLOGIC and our proposed NRETM model improve the performance over the T5-Large baseline model. Our NRETM model outperforms NEUROLOGIC by 5.3 CIDEr and 1.5 SPICE. This could be because NEUROLOGIC is an inference-only approach and is unable to change T5 internal neural representations to fully control the model to generate optimal sentences. We will add this result to our new version.

|         Model         | CIDEr | SPICE | Constraint (All) |
|:---------------------:|:-----:|:-----:|:----------:|
|        T5-Large       | 163.6 |  32.4 |    93.9    |
| T5-Large + NEUROLOGIC | 169.7 |  32.3 |    99.0    |
|    T5-Large + NRETM   | 175.0 |  33.8 |    99.4    |

**Encoder Update Baseline**

We implement the natural baseline that **Reviewer HmFk** mentioned and compare it with our proposed NRETM models in the Controlled ROC Story generation and commonsense generation task (see table below). This ablation study shows that updating the rule execution progress in the encoder contributes little to improve the constraints success ratio. In the Controlled ROC Story generation task, compared to the T5 baseline, this updating encoder baseline achieves a similar ROUGE-L score and slightly low CSR (roughly 1\% difference). In the commonsense generation task, the updating encoder baseline improves CIDEr score (from 159.5 to 163.2) and constraint success rate (from 94\% to 95.1\%). This shows that simply extracting rule execution intermediate values is not enough. This could be because the encoder that encodes the rule execution intermediate values cannot effectively broadcast this information into text decoders. In addition, our relative-position style method allows these intermediate values to be well aligned with the corresponding predicates in the encoder, allowing accurate control over text generation output. We will add this result to our new version.

| Predicate Logic Constraint |      Model      |  R-L |  CSR |
|:--------------------------:|:---------------:|:----:|:----:|
|             (1)            |   T5-Baseline   | 33.1 | 98.7 |
|                            |  T5-Update-Enc  | 33.0 | 98.6 |
|                            | T5-NERTM (Ours) | 33.3 | 98.8 |
|             (2)            |   T5-Baseline   | 49.6 | 97.0 |
|                            |  T5-Update-Enc  | 49.6 | 96.9 |
|                            | T5-NERTM (Ours) | 49.5 | 97.4 |
|             (3)            |   T5-Baseline   | 32.9 | 18.5 |
|                            |  T5-Update-Enc  | 32.9 | 16.5 |
|                            | T5-NERTM (Ours) | 32.6 | 84.5 |
|             (4)            |   T5-Baseline   | 33.0 | 23.5 |
|                            |  T5-Update-Enc  | 33.1 | 21.2 |
|                            | T5-NERTM (Ours) | 33.1 | 49.4 |
|             (5)            |   T5-Baseline   | 32.9 | 11.0 |
|                            |  T5-Update-Enc  | 33.2 |  7.9 |
|                            | T5-NERTM (Ours) | 33.0 | 35.9 |

|       Model      | CIDEr | SPICE | Constraint (All) |
|:----------------:|:-----:|:-----:|:----------:|
| T5-Base-Baseline | 159.5 |  31.9 |    94.0    |
|   T5-Update-Enc  | 163.2 |  32.1 |    95.1    |
|       Ours       | 169.8 |  32.7 |    99.3    |

Note that updating encoder content at each generation step results in significantly increased training time because one needs to re-compute the encoder content at each time step. It also stops the parallel training for transformer-based text decoders. In addition, not all prior knowledge can be expressed via changing encoder content. For example, in document-level NMT, our proposed NRETM model focuses on different sentences at different generation steps. However, removing these translated sentences from the encoder could reduce useful context information to the decoders (i.e., translated sentences are also useful context for the models to make decisions).



[1]  Ximing Lu, Peter West, Rowan Zellers, Ronan Le Bras, Chandra Bhagavatula, and Yejin Choi.  Neuro-logic decoding:(un) supervised neural text generation with predicate logic constraints.arXiv preprintarXiv:2010.12884, 2020

---

### Decision · Program_Chairs · 2021-09-27

**Decision:**

Accept (Poster)

**Comment:**

This paper proposes a technique for including global constraints during decoding, and letting these be softly violated during decoding. Although the technique seems to be reasonable (there were a few concerns about the terseness of the approach), and reasonably evaluated, the problem of constrained optimisation is one of (if not the most) widely studied problem in optimisation, and indeed, constrained optimisation has come up numerous times even in "decoding problems" (e.g., frameworks for decoding like dual decomposition, Rush and Collins, 2012, and lagrangian relaxation, Martins et al., 2012 et passim), and using first order predicate logic to express constraints has a long history (the constrained conditional models of Rizzolo and Roth, 2007, and Markov Logic Networks of Poon and Domingos). I would encourage the authors to explicate how this work relates to other work in constrained optimisation in the context of decoding algorithms, and in particular, to be more precise in the motivation and nature of the constraints used in the experiments (for instance, are they designed on a validation set and then tested on a held-out test set?).